# A mixed methods evaluation of a pilot open trial of a mentor-guided digital intervention for youth anxiety

Emma C. Wolfe[1]*, Alexandra Werntz[2], Audrey Michel[1], Yiyang Zhang[1], Mark Rucker[3], Mehdi Boukhechba[3], Laura E. Barnes[3], Jean E. Rhodes[2], Bethany A. Teachman[1]

**1** Department of Psychology, University of Virginia, Charlottesville, Virginia, United States of America, **2** Center for Evidence-Based Mentoring, University of Massachusetts, Boston, Massachusetts, United States of America, **3** Department of Engineering, University of Virginia, Charlottesville, Virginia, United States of America

* rpu3zk@virginia.edu

## Abstract

Digital mental health interventions (DMHIs), such as cognitive bias modification for interpretations (CBM-I), offer promise for increasing access to anxiety treatment among underserved adolescents, but data regarding their efficacy are mixed. Para-professionals and other caring adults in youth's lives, such as non-parental adult mentors, may be able to support the use of DMHIs and increase teen engagement. The present mixed methods evaluation of a pilot open trial tested the feasibility, acceptability, and preliminary efficacy of implementing MindTrails Teen (an app-based, youth-adapted version of the web-based MindTrails CBM-I intervention) within mentor/mentee dyads. Thirty participants (composed of 15 dyads) participated in remote data collection for 5 weeks. A subset of participants (n = 7 mentors; n = 7 mentees) also provided qualitative feedback. Intervention outcomes (change in anxiety symptoms, and positive and negative interpretation bias), feasibility, and acceptability were assessed via a mix of qualitative interviews, quantitative change in questionnaire scores, and program completion and fidelity metrics. Outcomes were compared to pre-registered benchmarks. Large effect sizes were observed for changes in anxiety among youth. Small to medium effects were observed for change in positive interpretation bias, and no change was found for negative interpretation bias. Intervention outcomes should be considered with caution given very low internal consistency of the interpretation bias measure and the lack of a control comparison group. Acceptability of the intervention was rated positively by mentors and youth. Feasibility benchmarks were met for mentors but not for youth. Qualitative feedback indicated mentors perceived the app as helpful to their mentees, found that it either improved or did not affect their relationship, but also identified implementation challenges. Youth overall perceived the app as helpful but identified barriers to engagement.

which permits unrestricted use, distribution, and reproduction in any medium, provided the original author and source are credited.

**Data availability statement:** The data used in this submission will be accessible on the Open Science Framework (quantitative data: https://osf.io/bqc2e/files/osfstorage; qualitative data: https://osf.io/nvukg/files/osfstorage) pending acceptance of this manuscript for publication.

**Funding:** This research was directly funded by a grant awarded to B.T. (NIMH Smart and Connected Health 1R01MH132138-01) and a grant awarded to A.W. (AIM Youth Mental Health (no grant number). The following two grants paid for indirect study costs (app development): President and Provost's Fund for Institutionally Related Research (no grant number); Center for Evidence-Based Mentoring postdoctoral research award (no grant number). J. R. was the director and A.W. the associate director of the Center for Evidence-Based Mentoring at the time of study data collection, analysis, and preparation of the manuscript. J.R. and A.W. assisted with preparation of the manuscript. Other than this, the funders had no role in study design, data collection and analysis, decision to publish, or preparation of the manuscript.

**Competing interests:** The authors have declared that no competing interests exist.

## Author summary

Youth anxiety is a serious public health concern, and digital mental health interventions (DMHIs) may provide low-cost, easy-to-access support. However, it can be challenging to reach youth with these interventions, and even more challenging to keep them engaged. Mentors, such as those in Big Brothers Big Sisters or related organizations, may be able to increase youth's use of such interventions and provide support and accountability. Following a 5-week pilot study of the cognitive-bias modification app MindTrails Teen, guided by mentors, youth reported being less anxious and feeling closer to their mentors. These outcomes should be considered in light of the small sample size and the absence of a control group. Youth and mentors reported that they overall liked the app and found the delivery style acceptable. Further research should test MindTrails Teen in a large comparative trial and more reliably test its impact on cognitive markers of anxiety.

## Introduction

Anxiety disorders are highly prevalent among adolescents. Approximately 15–20% of U.S. teenagers meet criteria for at least one anxiety disorder, and there is evidence of increased rates of youth anxiety since the onset of the COVID-19 pandemic [1–3]. Despite this, only one in five anxious adolescents report any lifetime mental health-care service use, and reported use is even lower for low-income teenagers and youth from historically marginalized groups (e.g., Black, Latinx, and LGBTQ youth [4–7]). Digitally delivered (e.g., internet or app-based) interventions are promising tools for increasing access to inexpensive and efficacious anxiety interventions among adolescents [8,9]. In many ways, adolescents present an ideal target population for digital interventions, given the ubiquity of smartphone use among teenagers (95% of teenagers in the U.S. aged 12–17 report owning a smartphone as of 2023 [10]). Digital interventions are feasible and acceptable for adolescents with a range of different anxiety disorders [8,9,11] and may be especially appealing to socially anxious teenagers, some of whom report preferences for technology-delivered treatments over face-to-face therapy [12].

One promising digital intervention for youth anxiety is digital cognitive bias modification for interpretations (CBM-I). Anxious youth tend to interpret ambiguous cues in their environment as threatening and respond with stress and/or avoidance, reinforcing anxious thinking as necessary [13]. CBM-I works by shifting these negative interpretations to be more positive, benign, or resilience focused, giving individuals repeated practice considering non-threatening explanations for ambiguous scenarios, which is intended to increase cognitive flexibility [13]. Digital CBM-I has been tested among adults with mostly favorable emotional and behavioral outcomes [14–17], but fewer studies have examined its efficacy in youth, and these findings are often mixed [18]. While some studies find improvements in anxiety symptoms among youth

[19–21], others report no change [22–24]. Conversely, nearly all studies of digital CBM-I for youth have found adaptive changes in interpretation bias, even when change in anxiety is not detected [22–24]. This suggests that, although digital CBM-I typically succeeds in activating a core change mechanism among youth (interpretation bias), this change may not always be sufficient to prompt clinically significant symptom change. More research is needed to understand how best to adapt and implement CBM-I for youth to promote symptom change.

MindTrails is a CBM-I intervention (available in both web and app formats) that has successfully reduced anxiety in adults across multiple trials [14–16], but has not yet been adapted or tested among youth. Several features of MindTrails suggest it may be well suited to target youth anxiety: one example is the brief nature of the intervention (15–20 minutes sessions), which may mitigate common barriers to engagement such as competing time demands and short attention spans [25,26]. Despite the efficacy of DMHIs such as MindTrails, engaging users remains a challenge. Evidence suggests that most teens who download a mental health app only open it a few times, and adherence varies widely (10%–94% in recent meta-analyses [27,26]). Human support has presented a replicable method of increasing engagement among users of DMHIs, but the shortage of providers hinders scalability [28–29]. Although MindTrails has primarily functioned as a fully self-guided intervention, attempts to increase efficacy and reduce attrition via human support have produced mixed results. In one study, low-intensity phone coaching was provided to a subset of adult trait anxious users of the MindTrails web platform, but approximately half of those assigned to the coaching condition never responded to attempts to schedule an initial phone call with their coach [30]. It remains unclear what impact human support might have among youth users with anxiety, especially if they have a preexisting relationship with the support person (vs. the stranger used in the prior MindTrails adult coach study).

Paraprofessionals (e.g., bachelor's level coaches, community leaders, teachers, mentors, etc.) present a promising avenue of human support for DMHIs, and—when properly trained and supervised—can be as effective as professional providers in delivering certain mental health services [31]. One candidate paraprofessional group for coaching youth in DMHIs is non-parent adult mentors. Youth mentoring programs are exceptionally well-positioned to support DMHIs. The basic contours of formal mentoring relationships follow those of professional helping relationships (e.g., often meeting once a week in mostly one-on-one relationships), and many youth mentees present with acute symptoms of anxiety, depression, and social, emotional, and behavioral struggles [32]. In fact, mental health concerns are often what prompt parent and teacher referrals, particularly among Black caregivers who may perceive professional treatment as stigmatizing or less culturally congruent [33]. This suggests a need for resources that mentors can use to address mentees' mental health needs in a structured, evidence-based format without requiring mentors to be experts in mental health treatment [34,35].

Despite its potential to address mental health needs, most mentoring produces only small effects on youth outcomes, in part because they tend to take a nonspecific, relationship-focused approach [36]. When mentoring relationships are focused on supporting youth in reaching a specific goal, the effect sizes are nearly double those of mentoring relationships that are friendship-based [37]. In a large demonstration project examining over 30 programs' outcomes, results revealed that striking a balance between supporting youth in accomplishing goals and building a strong relationship – in contrast to focusing on only one or the other – was associated with decreased depressive symptoms and emotional challenges among youth [38]. As a result, researchers and practitioners are exploring how mentors can provide targeted mental health support to mentees within the context of caring mentoring relationships. For youth struggling with mental health concerns, mentors can offer targeted support by providing supportive accountability to mentees using DMHIs [39]. Under this model, mentees learn evidence-based mental health skills (such as cognitive restructuring and challenging interpretation biases) through mental health apps, and mentors provide support, encouragement, and trouble-shooting.

By leveraging mentors as coaches, there is also a greater likelihood of established trust between the coach and the user, overcoming a common barrier to rapport and accountability in coached DMHIs [40,41]. Given the limited research and mixed findings associated with digital CBM-I for youth, we pilot tested a novel 5-week mentor-guided, app-based

CBM-I intervention (MindTrails Teen) for anxious teenagers participating in formal or informal mentoring programs. To our knowledge, few studies have examined the role of human support in increasing adherence to and engagement with CBM-I for youth anxiety, and no prior studies have used pre-existing mentors as human support for a digital intervention. Prior to the present study, we adapted a version of our existing online, adult-focused CBM-I intervention (MindTrails) into an app for a youth audience through iterative design and content changes supported by feedback from three youth consultants. This study represents the first pilot open trial of MindTrails Teen among youth participants. To this end, intervention and implementation framework feasibility and acceptability, target engagement (e.g., positive and negative interpretations) and preliminary anxiety symptom outcomes among youth are considered primary outcomes. We also present qualitative feedback collected from a subset of participants (more information about this data collection and associated research questions can be found on the Open Science Framework; preregistration for qualitative analyses: https://osf.io/pkbd5). Our analysis plan applies the mixed methods strategy of concurrent triangulation, using qualitative and quantitative methods to approach the same set of questions in service of a richer perspective [42]. Increasingly, this type of mixed methods approach is recommended for pilot intervention trials designed to examine preliminary intervention efficacy alongside feasibility and acceptability [43]. The quantitative data provide data-driven benchmarks, while the richness of the qualitative data allows us to probe participant experiences and perspectives more deeply. For example, although preliminary quantitative data in a small open trial might broadly indicate change in anxiety over time, it cannot explore which features of the intervention or implementation model participants believe drove this change. Though it remains impossible to establish a causal relationship in such a trial, the qualitative data offers critical insights to contextualize quantitative findings and more accurately inform future work. To clarify the complementary nature of our quantitative and qualitative results, we have added Table A in the Supplementary Materials (S1 Text) based on previous work by Aschbrenner and colleagues (2022) [43]. To briefly summarize, we used quantitative benchmarks as our primary outcome variables for feasibility, acceptability, and effectiveness, and designed our interview guide to elicit richer detail from participants on each metric.

We propose two sets of hypotheses; one set for anxious youth participants and one tied to inclusion of their mentors (preregistration for quantitative analyses: https://osf.io/rtcqk). First, we hypothesize that MindTrails Teen will be feasible and acceptable for anxious youth. We additionally hypothesize that MindTrails Teen will be accompanied by increased positive and decreased negative interpretation bias among anxious youth, and that MindTrails Teen will be accompanied by reduced youth anxiety symptoms. Finally, we hypothesize that including mentors in MindTrails Teen to provide supportive accountability to youth will be feasible and acceptable to both mentors and youth mentees.

## Methods

### Participants

Participants were recruited from across the United States between the period of June 1st, 2022 and May 2nd, 2023. Participants were recruited in dyads, such that each adolescent was required to join with a mentor or vice versa. Mentors were eligible to participate if they were: a) 18 years or older, b) had a smartphone or tablet, and c) served as a formal (e.g., a member of Big Brothers Big Sisters or another official mentoring organization) or informal (e.g., youth pastor, soccer coach) mentor to a teenager aged 13–17. Mentors were ineligible if they were immediate nuclear family members of the teen as we were seeking to isolate the effects of non-parent adults for this pilot test. Youth were eligible for participation if they were: a) 13–17 years old, b) had a smartphone or tablet, c) had consent of a parent or guardian, d) were living in the United States, e) were participating in a formal or informal mentoring program, f) scored >5 (mild range of anxiety [44]) on the Generalized Anxiety Disorder Questionnaire 7 (GAD-7) or if they self-reported an anxiety disorder diagnosis. As a note, the category "youth" is variably defined across the DMHI literature. We selected this specific age group (13–17) to align with the developmental norms of teenagers in United States, from where our sample was recruited. U.S. minors

are legally defined as individuals below the age of 18, and most U.S. teenagers enter high school (e.g., 8th-12th grades) between the ages of 13 and 14. During this stage of development, most adolescents will experience key cognitive-emotional transitions including a desire for autonomy and self-reliance and the ability to engage in complex forms of meta-cognition [45,46]. The latter can lead to an increase in self-focused rumination and is associated with the development of anxiety disorders in older adolescents [47]. Given the dynamic nature of anxiety symptoms and triggers across different stages of youth development (e.g., likelihood that romantic partners and social media would be a source of anxiety), we chose to tailor the content of the app to be appropriate for adolescents in this developmental stage.

Due to financial and personnel restrictions, the team recruited a convenience sample. To reduce local bias and broaden reach as far as possible, recruitment took place via postings on social media (e.g., Twitter, Reddit, Facebook), placing fly-ers in local community centers and mentoring organizations, sending emails to the leadership of a broad range of national and local mentoring organizations across the U.S., and presenting at national mentoring conferences. Although the team originally targeted only formal mentoring programs, recruitment was expanded to include informal mentors (e.g., youth pastors, community leaders) to enhance representation and reach. Despite these efforts, recruitment within an abbrevi-ated timeline proved a significant challenge and the team ultimately recruited only 21 dyads (n = 42) out of our goal of 35 dyads (n = 70). Out of these 21 dyads, one was not eligible due to a subclinical youth GAD-7 score. Five completed the consent process and were given an app code but the youth never downloaded the app, leaving a final sample of 15 dyads (n = 30; 15 mentors and 15 teens) who completed the study. Complete demographic data were available for 28/30 partic-ipants, with one mentor and one youth declining to provide their demographic information. Of the mentors who provided demographic data, the sample was 40.21 years old on average ($SD$ = 8.96; range = 27–55), majority female (64%), White (93%), and non-Hispanic (93%). Of the teens who provided demographic data, the sample was 15.14 years old on aver-age ($SD$ = 1.01; range = 14–17), majority male (50%, although 34% did not identify with the gender binary), White (67%), and non-Hispanic (83%). For other sociodemographic information, see Table 1.

## Study procedure

All study procedures were approved by the University of Virginia Institutional Review Board prior to recruitment. Interested participants contacted the study team via email, and youth eligibility was confirmed via a Qualtrics survey. The study team then shared consent forms electronically via DocuSign to the mentor and parent/guardian and an assent form to either the youth or the parent to share with the youth. Once all three parties had consented or assented, the study team con-tacted the mentor to schedule a 30-minute training session focused on: 1) psychoeducation about anxiety and cognitive bias modification, and 2) how to provide supportive accountability to promote youth engagement with MindTrails Teen. An app download code was also shared with youth at this time. It is unknown whether mentors had any training or previous experience delivering digital or in-person psychological interventions, but none was reported to our study team during the duration of the pilot trial.

The study took place over five weeks, with an additional one-month follow-up assessment. Teens were instructed to complete at least two 15–20-minute sessions on the MindTrails-Teen app per week. Mentors were asked to meet with (or at minimum check in via email, phone, or text) with their mentees once a week. Mentors were informed about mentee progress via a weekly email from the research staff. Study staff could view youth progress and sessions completed on an internal dashboard. If the mentee had not completed any sessions during the prior week, mentors were asked by study staff to perform an additional check-in.

In addition to the youth mentee completing the initial Qualtrics eligibility survey, both mentors and youth completed Qualtrics surveys at baseline (pre-intervention), three weeks (mid-intervention), and five weeks (post-intervention) into the study. Mentees completed an additional follow-up Qualtrics survey one month following completion of the study. Each week, mentors were also sent a brief survey alongside their mentees' progress report asking about their contact with their mentee. Finally, at the end of the five-week primary study, both mentors and youth were invited to take part in a

**Table 1. Sociodemographic characteristics of the samples.**

| Baseline Characteristics | Teens | | Mentor | |
|---|---|---|---|---|
| | n | % | n | % |
| Gender | | | | |
| Female | 3 | 21.43 | 9 | 64.29 |
| Male | 7 | 50 | 5 | 35.71 |
| Other/nonbinary identity | 3 | 21.43 | 0 | 0 |
| Prefer Not to Answer | 1 | 7.14 | 0 | 0 |
| Race (Participants could select more than one category) | | | | |
| American Indian/Alaskan Native | 1 | 7.14 | 0 | 0 |
| Black/African Origin | 0 | 0 | 1 | 7.14 |
| White/European | 8 | 57.14 | 13 | 92.86 |
| More than one race | 3 | 21.43 | 0 | 0 |
| Other/Unknown | 2 | 14.29 | 0 | 0 |
| Ethnicity | | | | |
| Hispanic/Latino | 2 | 14.29 | 1 | 7.14 |
| Non-Hispanic/Latino | 12 | 85.71 | 13 | 92.86 |
| State of Residence | | | | |
| Iowa | 5 | 35.71 | 5 | 35.71 |
| Indiana | 1 | 7.14 | 1 | 7.14 |
| Massachusetts | 1 | 7.14 | 1 | 7.14 |
| North Carolina | 1 | 7.14 | 1 | 7.14 |
| Nebraska | 3 | 21.43 | 3 | 21.43 |
| Texas | 3 | 21.43 | 3 | 21.43 |
| Grade | | | | |
| 7th Grade | 2 | 14.29 | 0 | 0 |
| 8th Grade | 4 | 28.57 | 0 | 0 |
| 9th Grade | 3 | 21.43 | 0 | 0 |
| 10th Grade | 5 | 35.71 | 0 | 0 |
| Length of Mentoring Match | | | | |
| 1 years | 4 | 28.57 | 4 | 28.57 |
| 2 years | 5 | 35.71 | 5 | 35.71 |
| 3 years | 1 | 7.14 | 1 | 7.14 |
| 4 years | 4 | 28.57 | 4 | 28.57 |
| Type of Mentorship Program | | | | |
| Formal | 12 | 85.71 | 12 | 85.71 |
| Informal | 2 | 14.29 | 2 | 14.29 |

*Note:* One teen and one mentor did not provide demographic data, thus the demographic data reflects $n = 28/30$ participants.

qualitative interview to better understand participant perspectives on MindTrails Teen and the study procedures. Interviews took place over Zoom, were administered by a member of the study team, and were audio-recorded. At the end of the study, participants (both mentors and youth plus their parent/guardian) were emailed a debriefing form.

Participants were compensated with online gift cards from Tango, a site that allows recipients to choose where they can spend their money online. Youth received up to $90 for participating and mentors received up to $70 for participating.

## Materials and measures

This manuscript details only those measures used for the current analyses. See Table B in the Supplementary Materials (S1 Text) for a table containing the full list of measures administered within the trial and the measure administration schedule.

**Demographic and clinical characteristics. Demographics:** All participants were asked to complete a demographic questionnaire assessing age, gender, race, ethnicity, and state of residence. Mentors were asked about level of education and occupation, while youth were asked about school grade level.

**Mentorship Questions:** Mentees were asked if they had a mentor, if they were in a specific mentoring program, and how long they had been working with their current mentor. Mentors were asked if they were participating in a specific mentoring program, how long they had worked with their mentee, and if they worked on targeted goals with their mentee.

**Interpretation bias measures. Recognition Rating Task:** Target engagement (change in positive and negative interpretations) was measured via the Recognition Rating Task (modified from Mathews & Mackintosh, 2000) [48]. A modified version of the Recognition Rating Task was administered to youth at three study timepoints. This task was included to evaluate baseline interpretation bias and test if CBM-I training modified interpretation bias in expected directions. The first part of the Recognition Rating task asked youth to read several short stories, adapted to situations common to teenagers and pilot tested with teen consultants. The last word of each story was incomplete, and youth were asked to click on the missing letter to complete each word fragment. After correctly completing the word, the youth mentee was asked to answer a question about the story. For example, one scenario was titled "The Test". This story read: *"You have a final test in your math class next week. You know it will be challenging and cover a lot of material. You need a plan to [st_dy]"* Youth entered the letter u to complete the word *"study".* Youth were then asked if the test would be difficult (the correct answer was *"Yes").* In this first task, the correct answer did not resolve the anxiety-related ambiguity of the situation (i.e., how the student performed on the test) and was intended as a comprehension check. Next, youth were asked to read the titles of the nine ambiguous scenarios they had just seen. After each title, four sentences were presented. Two of these sentences were tied to interpreting the situational ambiguity, and two were "foils" that did not assign positive or negative meaning relevant to the anxiety-linked threat. Using "The Test" scenario as an example, one interpretation is negative and anxiety-related ("And you think that you will do badly") and one interpretation is positive and resilience-related ("And you think that you will do well"). The remaining two sentences were foils, meaning that they had positive or negative valence but did not relate to the anxiety-linked emotional ambiguity ("And you think about how much you like your teacher" – positive foil; "And it will take a long time to finish" – negative foil). Youth were asked to read these four sentences and rate them on a scale of 1–4 (1*: Very Different;* 2*: Different;* 3*: Similar;* 4*: Very Similar; Prefer Not to Answer)* based on how similar each sentence was to the original story that was read. They were then instructed to answer based on their understanding of the content of the story, rather than what they would think or do in this situation. The measure had poor internal consistency ($a = 0.35$ for positive interpretation bias items, $a = 0.48$ for negative interpretation bias items), suggesting its results should be interpreted with some caution.

**Mood measures. The Generalized Anxiety Disorder Questionnaire-7** (GAD-7 [43]) is a seven-item questionnaire that assesses frequency of anxiety and worry in the past two weeks using a four-point Likert scale from *not at all* (0) to *nearly every day* (3). Item ratings were summed to calculate a total anxiety severity, with higher scores reflecting greater severity. This measure was used exclusively at baseline to determine study eligibility.

**Self-Reported Anxiety Diagnosis** is a single-item measure administered at baseline that asked mentees if they have been diagnosed by a doctor or other healthcare professional with any anxiety disorder, including panic disorder, specific phobia, generalized anxiety disorder, separation anxiety disorder, social anxiety disorder, or agoraphobia. Responses were "*Yes*"; "*No*"; "*I don't know*" or "*Prefer not to answer*". It was included to more fully characterize the sample and function as a secondary inclusion criterion for patients whose anxiety might not be captured by the PHQ-4 anxiety subscale (also known as the GAD-2).

**The Patient Health Questionnaire-4** (PHQ-4) [49] is a four-item questionnaire that assesses frequency and severity of anxiety and depression in the past two weeks using a four-point Likert scale from *not at all* (0) to *nearly every day* (3). The first two items assess anxiety symptoms, and the second two items assess depression symptoms. Ratings for the first two items (also known as the Generalized Anxiety Disorder Scale-2 (GAD-2)) were averaged together and the average multiplied by two to create an overall anxiety score for each time point, with higher scores reflecting greater severity. Both the PHQ-4 and GAD-2 have good reliability [50]. This measure was originally selected for its good reliability and to minimize burden on youth participants.

**Acceptability. MindTrails Acceptability Questionnaires (MAQ) Mentor and Mentee Versions:** Acceptability was measured for youth via the MindTrails Acceptability Questionnaire (MAQ): Mentee Version, which is a seven-item self-report measure of youth opinions about the MindTrails Teen program delivered immediately post-intervention, following the conclusion of the study at Week 5. Acceptability was measured for mentors via the MindTrails Acceptability Questionnaire (MAQ): Mentor Version, which is a nine-item self-report measure of youth opinions about the MindTrails Teen program delivered immediately post-intervention, following the conclusion of the study at Week 5. Both measures were developed by the study team using modified measures of DMHI acceptability from other pilot feasibility trials [51,52]. Both versions use a five-point Likert response scale from "*Strongly agree*" (1) to "*Strongly disagree*" (5).

**Weekly Mentor Survey:** This investigator-designed survey was sent weekly to mentors to monitor program adherence. It contained four questions designed to capture if the mentor had met with their mentee that week, whether the meeting was in-person or virtual, and if and how the mentor and youth had discussed MindTrails Teen during the meeting. Mentors were also asked to write a short paragraph describing interactions with their mentee during the week.

**CBM-I Protocol.** All mentees were offered the same CBM-I training program over a five-week period. The training was delivered in brief, 10–20-minute doses designed to give users practice shifting negative interpretations and learning to think in flexible ways. Youth were able to unlock and complete a maximum of one session per day and two sessions per week for a total of 10 sessions across the five-week period. Once the youth enrolled with their QR code, they were prompted to begin the first session. The first two sessions were designed to be introductory. For details of these sessions, see the Supplementary Materials (S1 Text). The remaining eight sessions were split evenly between two different types of training sessions to have a variety of ways of engaging and to encourage cognitive flexibility. We labeled these two sessions "short" training sessions and "long" training sessions, given that youth completed a series of brief exercises in the short sessions and one long exercise in the long session.

The short training sessions were consistent with the typical presentation of MindTrails CBM-I in previous web and app-based versions [15,16]. In these sessions, youth were presented with a brief, ambiguous scenario followed by a word fragment which, when completed, would resolve the scenario either positively (70% of cases), resiliently (20% of cases) or negatively (10% of cases). In the "positive" case, the scenario would resolve in a good or desired outcome. In the "resilient" case, the scenario would have a mixed outcome. In the "negative" case, the scenario would resolve in a bad or undesired outcome. Youth were asked to read the scenario and then complete the word fragment. To personalize this experience, youth were asked prior to each session to select from a list of six stressor domains, generated with help from teen consultants (Domains: Academics, Physical Health, Home Life, Social Situations, Social Media, and General), and were presented with scenarios tied to this domain. The primary goal of these short scenarios was to give repeated practice in making positive or resilient interpretations of ambiguous scenarios to reduce the tendency of making rigid negative interpretations.

The long training sessions were a new addition to MindTrails based on work by Silverman, Fua, and Teachman (2023) [53] and focused on generating many different thoughts, feelings, and behaviors that might follow a single ambiguous situation. Instead of completing a word fragment in response to a scenario, as in the short-sessions, youth were asked to vividly imagine themselves in a stressful situation related to their chosen domain. After picturing themselves in the situation, youth were presented with five example thoughts, five example feelings, and five example behaviors they might

have had (in each case two were positive/adaptive, two were designed to build resilience, and one was negative/maladaptive). These were shown to demonstrate that many different thoughts, feelings, and behaviors could occur in response to ambiguous scenarios in order to encourage generative, flexible responding and perspective taking. For each of the three sections (thoughts, feelings, and behaviors), after youth were shown the five examples, they were asked to spend 30 seconds writing down their own thoughts, feelings, or behaviors in response to the imagined situation. They were then prompted to write down things they could tell themselves in a similar situation to help them handle feelings of anxiety. Thus, the aim of the long training session was to provide examples of the many different reactions a person could have to a stressful situation and encourage youth to consider how they might prioritize helpful reactions and manage negative reactions.

Each week, youth were instructed to complete a short training session in the first half of the week and a long training session in the second half of the week. Guided imagery exercises were presented in advance of each training session to prime youth for the imaginal component of the sessions and give them practice vividly imagining themselves in a scenario. See the Supplementary Materials (S1 Text) for descriptions of the guided imagery tasks and a detailed breakdown of which training sessions were assigned to each week.

**Qualitative interviews.** Both youth and mentors were invited to complete an interview following completion of the program and compensated an additional $20 for participating. Mentees provided feedback on the app/intervention and offered ideas to improve the aesthetics of the app, increase engagement and adherence, and more broadly increase usability for future teen users. Mentors provided feedback on the training they received and offered ideas to improve mentee engagement with and adherence to the program. For more information, see the interview guide provided in the Supplementary Materials (S1 Text).

**Feasibility outcomes.** *Intervention feasibility* among youth was measured in two ways. First, we assessed clinical deterioration (based on the assumption that if the program is not safe, then it is not feasible), comparing rates of anxiety score increases on the GAD-2 scale to benchmarks pulled from existing literature. Second, we assessed adherence to the MindTrails Teen protocol. For youth, adherence was captured based on percentage of program completion (i.e., % of 10 sessions completed).

*Implementation framework feasibility* was measured in two ways. First, we evaluated mentor and youth adherence to the meeting schedule. This was captured by the number of times a "Yes" response was given to Question 1 of the Weekly Mentor Survey, which asked: "Did you meet with your mentee this week?" Second, we evaluated mentor and youth discussion of MindTrails Teen. This was captured by the number of times a "Yes" response was given to Question 3 of the Weekly Mentor Survey, which asked: "If you did meet, did you discuss your mentee's use of MindTrails?" To avoid conflating missing responses (incomplete surveys) with poor adherence tied to the mentor's role in supporting the mentee's use of the program, we only assessed the above markers of feasibility based on completed questionnaires.

**Acceptability outcomes.** Prior to conducting analyses, we identified three items from the MAQ: Mentee version as primary measures of *intervention acceptability* among youth, three items from the MAQ: Mentee version as primary measures of *implementation framework acceptability* among youth, and three items from the MAQ: Mentor version as primary measures of *implementation framework acceptability* among mentors and youth. See Table 2 for more information on specific acceptability questions.

## Data analysis plan

**Quantitative analysis. Feasibility Benchmarks:** To test our hypothesis that the MindTrails intervention would be feasible for youth, we examined rates of clinical deterioration and protocol adherence. See Table 2 for specific feasibility benchmarks.

**Acceptability Benchmarks:** To test our hypothesis that the MindTrails intervention would be acceptable for youth, we examined descriptive data for the three items established *a priori* as acceptability markers and compared these results

**Table 2. Feasibility benchmark descriptions, rationale, and results.**

| Benchmark | Rationale | Result | Benchmark Met? |
|---|---|---|---|
| **Intervention Feasibility (Mentee)** | | | |
| Clinical deterioration<br>• Less than 20% of participants (3/15) will experience an increase in symptoms on the anxiety subscale of the GAD-2 > 50% above their pre-intervention score | Prior studies evaluating the standard English version of MindTrails have used similar criteria to assess for iatrogenic effects, with results indicating very few negative effects for participants [14,15]. For example, in Ji et al., only 18 participants (out of the 807 randomized to a training condition) experienced an increase in symptoms on the OASIS greater than 50% above their baseline score. | Clinical deterioration<br>• 0/15 participants experienced an increase in symptoms on the anxiety subscale of the GAD-2 > 50% above their pre-intervention score | **Yes** |
| Protocol Adherence<br>• At least 46% of enrolled mentees (7/15) will complete ½ of assigned CBM-I training sessions (5/10 sessions).<br>• At least 33% of enrolled mentees (5/15) will complete all assigned CBM-I training sessions (10/10 sessions) | Based on our literature review of feasibility of digital CBM-I for anxious youth, youth completed around half or slightly less than half of their assigned CBM-I sessions (Participants completed an average of 5.85/8 sessions [24], 49% completed at least half of all assigned sessions [20], and around a third completed all sessions (38% [22], 37% [20]). Notably, all three of these trials included >100 participants and were compensated, and therefore rates of completion may not reflect real-world implementation of digital CBM-I for anxious youth. All three trials were also unguided (e.g., children/adolescents completed their CBM-I independently): the broader literature on DMHIs for youth is mixed as to whether human support improves adherence to DMH interventions but at least one meta-analysis has suggested higher adherence with human support [8]. Given the small, compensated, guided nature of our trial, we expect adherence and enrollment among youth to follow similar trends to the existing literature. | Protocol Adherence<br>• 40% of enrolled mentees (6/15) completed ½ of assigned CBM-I training sessions (5/10 sessions).<br>• 20% of enrolled mentees (3/15) completed all assigned CBM-I training sessions (10/10 sessions) | **No** |
| **Implementation Framework Feasibility (Mentor & Mentee)** | | | |
| Protocol Adherence<br>• At least 53% of mentors (8/15) will complete ½ of all check-ins.<br>• At least 53% of mentors (8/15) will discuss MindTrails with their mentees in ½ of all completed check-ins. | Given that this is a real-world implementation study, we do not expect that all mentors will maintain perfect fidelity to the protocol, and we have set these benchmarks based on what we would expect as the minimum necessary contacts and program check-ins given real-world implementation hurdles. There is not a rich literature base to guide the specific percentages, but we would hope that at least half the people who engage with the program as a mentor will do so in a way that allows for a substantive 'dose' of the mentorship component of the MindTrails Teen program. | Protocol Adherence<br>• 60% of mentors (9/15) completed ½ of all check-ins.<br>• 60% of mentors (9/15) discussed MindTrails with their mentees in ½ of all completed check-ins. | **Yes** |

to benchmarks pulled from the literature (See Table 3). To test our hypothesis that the intervention implementation model would be acceptable to youth and mentors, respectively, we examined descriptive data for three items each that were established *a priori* as implementation acceptability markers and compared these results to benchmarks pulled from the literature. See Table 3 for acceptability benchmarks.

**Intervention Outcomes:** The preliminary nature of the study and small sample size meant we were not powered to detect statistical significance using traditional hypothesis testing. Therefore, to test for change in intervention outcomes (anxiety symptoms and positive and negative interpretation bias, which are our measures of target engagement) we calculated Cohen's *d* effect sizes with confidence intervals separately for the GAD-2, positive interpretation bias, and negative interpretation bias. We report confidence intervals for the sake of transparency, but do not use these to interpret statistical significance given the small sample size. Based on prior work using within-subjects design with a small sample size [57] and consultation with an internal statistics consultant (C. Ford, personal communication, August 12, 2024), we calculated $d_{av}$, using the average standard deviation of the pre and post repeated measures as a standardizer:

**Table 3.** Acceptability benchmark descriptions, rationales and results.

**Intervention Acceptability (MAQ: Mentee Version)**

*Acceptability of MindTrails Intervention*

| Benchmark | Rationale | Result | Benchmark Met? |
|---|---|---|---|
| 1. "I enjoyed using the MindTrails app." <br>• Mean ≥ 3 (between "Neither agree nor disagree" and "Agree") | Mentee Intervention Acceptability Items #1–3: Youth DMHIs tend to rate high on acceptability. Studies using similar 5-point Likert scales to measure intervention acceptability tend to report mean ratings between 3 and 4 [54,55]. | 1. "I enjoyed using the MindTrails app." <br>• Mean = 3.56 (SD = 1.59) | **Yes** |
| 2. "I would recommend MindTrails to other mentees or teens." <br>• Mean ≥ 3 (between "Neither agree nor disagree" and "Agree") | | 2. "I would recommend MindTrails to other mentees or teens." <br>• Mean = 4.11 (SD = 1.17) | **Yes** |
| 3. "MindTrails changed the way I think during anxiety-provoking situations." <br>• Mean ≥ 3 (between "Neither agree nor disagree" and "Agree") | | 3. "MindTrails changed the way I think during anxiety-provoking situations." <br>• Mean = 3.56 (SD = 0.73) | **Yes** |

**Implementation Framework Acceptability (MAQ: Mentee Version)**

*Acceptability of Mentorship Framework for Mentee*

| Benchmark | Rationale | Result | Benchmark Met? |
|---|---|---|---|
| 1. "I enjoyed working with my mentor to apply the skills I learned in MindTrails." <br>• Mean ≥ 3 (between "Neither agree nor disagree" and "Agree") | Mentee Implementation Acceptability Items #1–3: Research on implementation framework acceptability among youth is limited and often not distinct from intervention acceptability. Therefore, our benchmark is based on what the research team views as the minimum necessary score to consider the implementation model acceptable for broader dissemination among mentor/mentee dyads. A mean score of <3 would indicate overall youth dissatisfaction with the implementation framework. | 1. "I enjoyed working with my mentor to apply the skills I learned in MindTrails." <br>• Mean = 4.22 (SD = 0.67) | **Yes** |
| 2. "I would recommend using MindTrails with a mentor to other mentees or teens." <br>• Mean ≥ 3 (between "Neither agree nor disagree" and "Agree") | | 2. "I would recommend using MindTrails with a mentor to other mentees or teens." <br>• Mean = 3.89 (SD = 1.17) | **Yes** |
| 3. "It was helpful to have my mentor support my use of MindTrails." <br>• Mean ≥ 3 (between "Neither agree nor disagree" and "Agree") | | 3. "It was helpful to have my mentor support my use of MindTrails." <br>• Mean = 4.22 (SD = 0.83) | **Yes** |

**Implementation Framework Acceptability (MAQ: Mentor Version)**

*Acceptability of Mentorship Framework for Mentor*

| Benchmark | Rationale | Result | Benchmark Met? |
|---|---|---|---|
| 1. "I enjoyed participating in this project." <br>• Mean ≥ 3 (between "Neither agree nor disagree" and "Agree") | Mentor Implementation Acceptability Items #1–3: Research on implementation framework acceptability among mentors and/or coaches is limited and often not distinct from intervention acceptability. Therefore, our benchmark is based on what the research team views as the minimum necessary score to consider the implementation model acceptable for broader dissemination among mentor/mentee dyads. A mean score <3 would indicate overall mentor dissatisfaction with the implementation framework. | 1. "I enjoyed participating in this project." <br>• Mean = 4.21 (SD = 0.80) | **Yes** |
| 2. "I felt prepared to help my mentee use MindTrails." <br>• Mean ≥ 3 (between "Neither agree nor disagree" and "Agree") | | 2. "I felt prepared to help my mentee use MindTrails." <br>• Mean = 3.87 (SD = 0.99) | **Yes** |
| 3. "I would recommend MindTrails with mentor assistance to other mentees." <br>• Mean ≥ 3 (between "Neither agree nor disagree" and "Agree") | | 3. "I would recommend MindTrails with mentor assistance to other mentees." <br>• Mean = 4.40 (SD = 0.83) | **Yes** |

**Intervention Outcomes**

| | | | |
|---|---|---|---|
| Pre- to post-intervention change in target engagement: <br>• Negative interpretation bias score Hedge's g ≥ .2 | All studies in our literature review examining multi-session digital interventions found at least a small effect of the intervention on changes in interpretation bias [19,20,22,24]. The only studies that did not produce an impact on bias were single session interventions [23,56]. This reflects trends in the broader literature on CBM-I among anxious youth in which CBM-I has small to moderate effects on positive and negative interpretations [56]. | Pre- to post-intervention change in target engagement: <br>• PP: Hedge's g = -0.12 <br>• ITT: Hedge's g = 0.15 | **No** |
| Pre- to post-intervention change in target engagement: <br>• Positive interpretation bias score Hedge's g ≥ .2 | | Pre- to post-intervention change in target engagement: <br>• PP: Hedge's g = 0.31 <br>• ITT: Hedge's g = 0.60 | **Yes** |

*(Continued)*

**Table 3.** (Continued)

| Intervention Acceptability (MAQ: Mentee Version) | | | |
|---|---|---|---|
| *Acceptability of MindTrails Intervention* | | | |
| **Benchmark** | **Rationale** | **Result** | **Benchmark Met?** |
| Pre- to post-intervention change in anxiety symptoms:<br>• GAD-2 anxiety subscale score Hedge's g ≥ .2 | Anxiety outcomes among youth using digital CBM-I interventions are mixed. Some suggest a small to moderate effect of CBM-I on anxiety [19,20] while others either find no effect of CBM-I on anxiety or a significant main effect of time such that both the control group and CBM-I group show changes in anxiety [22–24].While we anticipate that change in anxiety will be a smaller effect than change in interpretation bias, we expect at least a small anxiety reduction effect based on our prior MindTrails outcomes in adults and given the predicted positive impact of mentorship on intervention adherence and engagement. | Pre- to post-intervention change in anxiety symptoms:<br>• PP: Hedge's $g$ = 1.16<br>• ITT: Hedge's $g$ = 0.88 | **Yes** |

$$\text{Cohen's } d_{av} = M_{diff}/((SD_1 + SD_2)/2)$$

We then applied Hedge's $g$ correction, which gives a less biased effect size for small sample sizes ($n < 20$) [58]:

$$\text{Hedge's } g_{av} = \text{Cohen's } d_{av} \times (1 - (3/(4(n_1 + n_2) - 9)))$$

We then compared these results to effect size benchmarks pulled from the literature (see Table 4). To inform future trials and better understand the response pattern, we also calculated percent change in scores from pre-intervention to post-intervention for each participant and on average, but we did not set a benchmark for this descriptive analysis. In the case of missing data for post-intervention (as was the case for two mentees), mid-intervention (e.g., week 3) scores were carried forward. Otherwise, all comparisons were between baseline (e.g., week 0) and post-intervention (e.g., week 5) scores.

Given that this is a pilot open trial, we did not conduct pre-trial power analyses to determine sample size. Based on previous pilot trials of similar scope, we initially set our estimated sample size at N = 35 mentor/mentee dyads. However, due to recruitment challenges and an abbreviated timeline, we ended up closing recruitment at N = 15 mentor/mentee dyads, of which N = 7 mentees and N = 7 mentors consented to post-intervention qualitative interviews. This sample size, though not ideal, is consistent with or close to other pilot studies reported in mixed methods literature assessing feasibility and acceptability of DMHIs for youth [21,59–61]. Furthermore, due to participant non-completion of surveys across both the midpoint and endpoint time intervals, we analyzed data from two different sub-samples. For each outcome, the intent to treat (ITT) sample aggregates all data available at each pre and post timepoint regardless of full survey completion, and per protocol (PP) sample examines only those participants who have two available data points for within-person pre-post analysis.

**Qualitative analysis.** Audio recordings of qualitative interviews were hand-transcribed by three undergraduate research assistants (YZ, EC, AM). The lead author (EW) supervised this work and read through all transcripts, correcting them against the audio recordings. After all interview transcripts were reviewed, the research team discussed possible themes until a consensus was reached and two initial codebooks were generated. Following this initial code generation, an inductive approach was applied, adding and deleting codes to best fit the data throughout the coding process [62]. Interviews were coded using Delve qualitative data analysis software [63] and coders were trained via intercoder reliability tests (ICR). Intercoder reliability was measured using Krippendorff's alpha, a measure of reliability in content analysis [64]. Once an ICR of .80 was reached, two people from the team

**Table 4. Intervention outcomes benchmark descriptions, rationales and results.**

**Intervention Outcomes**

| Benchmark | Rationale | Result | Benchmark Met? |
|---|---|---|---|
| Pre- to post-intervention change in target engagement:<br>• Negative interpretation bias score Hedge's g ≥ .2 | All studies in our literature review examining multi-session digital interventions found at least a small effect of the intervention on changes in interpretation bias [19,20,22,24]. The only studies that did not produce an impact on bias were single session interventions [23,52]. This reflects trends in the broader literature on CBM-I among anxious youth in which CBM-I has small to moderate effects on positive and negative interpretations [56]. | Pre- to post-intervention change in target engagement:<br>• PP: Hedge's $g = -0.12$<br>• ITT: Hedge's $g = 0.15$ | No |
| Pre- to post-intervention change in target engagement:<br>• Positive interpretation bias score Hedge's g ≥ .2 | | Pre- to post-intervention change in target engagement:<br>• PP: Hedge's $g = 0.31$<br>• ITT: Hedge's $g = 0.60$ | Yes |
| Pre- to post-intervention change in anxiety symptoms:<br>• GAD-2 anxiety subscale score Hedge's g ≥ .2 | Anxiety outcomes among youth using digital CBM-I interventions are mixed. Some suggest a small to moderate effect of CBM-I on anxiety [19,20] while others either find no effect of CBM-I on anxiety or a significant main effect of time such that both the control group and CBM-I group show changes in anxiety [22–24]. While we anticipate that change in anxiety will be a smaller effect than change in interpretation bias, we expect at least a small anxiety reduction effect based on our prior MindTrails outcomes in adults and given the predicted positive impact of mentorship on intervention adherence and engagement. | Pre- to post-intervention change in anxiety symptoms:<br>• PP: Hedge's $g = 1.16$<br>• ITT: Hedge's $g = 0.88$ | Yes |

were assigned to independently code each of the remaining interviews, and coding discrepancies were discussed among the research team in weekly meetings. For these meetings, nominal group consensus was applied, in which the first author served as the moderator to help settle disputes in coding. The team used an inductive coding approach, adding and deleting codes to best fit the data throughout the coding process [62]. After each transcript was discussed, a "final" version was uploaded in Delve. The final mentee codebook consisted of 17 primary codes organized into 5 themes, and the final mentor codebook consisted of 22 primary codes organized into 8 themes (See OSF database for codebooks: https://osf.io/pkbd5)

Following the initial coding process, the team engaged in thematic mapping of both codebooks to visualize the codes and their associated relationships (see Figs A and B in the Supplementary Materials (S1 Text) for the preliminary thematic maps, Figs 4 and 5 for the final thematic maps). Codes with low saturation (e.g., endorsed by 1/7 participants or less) were collapsed into broader categories, and several codes with higher saturation were separated into new themes. At the conclusion of this process, the team was left with 7 mentee themes containing a total of 18 primary codes (Table 5 and and Figs 4 and 5), and 7 mentor themes containing a total of 14 primary codes (Table 6). These procedures are consistent with those conducted in previous qualitative evaluation of digital mental health intervention and implementation [65,66].

## Quantitative results

Pre-set benchmarks and associated results are presented in Table 4. Table 7 reports percent change experienced by each participant with at least two available data points (Per Protocol sample). See Supplementary Materials (S1 Text) for minor deviations from the quantitative preregistration.

**Feasibility (Hypotheses 1A & 2C).** Regarding *intervention feasibility* for youth, benchmarks were met for clinical deterioration, with no participants displaying a GAD-2 anxiety subscale score increase >25% over the course of the 5-week intervention period, though one participant experienced a score increase of exactly 25% (score increase from 4

**Table 5. Main themes identified and frequency of codes among mentees.**

| Theme | Code | Number Mentors Endorsed (out of N=7) |
|---|---|---|
| **Theme 1:** Facilitators of Engagement | Components of the intervention that increased engagement | 5 |
| | Components of the app that increased engagement | 4 |
| **Theme 2:** Barriers to Engagement | Components of the intervention that decreased engagement | 6 |
| | Individual characteristics that decreased engagement | 4 |
| | Components of the app that decreased engagement | 3 |
| | External factors that decreased engagement | 2 |
| **Theme 3:** Suggestions for Improving Engagement | -- | 7 |
| **Theme 4:** Change in Mental Health due to MindTrails Teen | Positive change | 4 |
| | No change | -- |
| | Negative change | -- |
| **Theme 5:** Description of MindTrails Teen | Positive description | 4 |
| | Neutral description | 4 |
| | Negative description | 1 |
| **Theme 6:** Perceived Value of Mentorship while using MindTrails Teen | Positive impact | 5 |
| | Neutral/no impact | 1 |
| | Negative impact | -- |
| **Theme 7:** Change in mentee/mentor relationship due to MindTrails Teen | Positive change | 3 |
| | Neutral/no change | 2 |
| | Negative change | -- |

**Table 6. Main themes identified and frequency of codes among mentors.**

| Themes | Code | Total Endorsements Total (N=7) |
|---|---|---|
| **Theme 1:** Perceived Value of Mentorship | Useful | 3 |
| | Not useful | 1 |
| **Theme 2:** Change in mentor/mentee relationship due to MindTrails Teen | Positive change | 4 |
| | Neutral/no change | 2 |
| | Negative change | 1 |
| **Theme 3:** Mentorship Meetings | Incorporation of MindTrails in meetings | 6 |
| **Theme 4:** Benefit of MindTrails Teen to Mentee Mental Health | Perceived changes in anxiety levels of mentee | 5 |
| **Theme 5:** Barriers to Engagement | External factors that decreased engagement | 3 |
| | Individual characteristics that decreased engagement | 2 |
| | Components of the app that decreased engagement | 2 |
| **Theme 6:** Suggestions for program improvement | Suggestions for improving mentor experience | 5 |
| | Implementation considerations | 2 |
| **Theme 7:** Mentor Experience of MindTrails Teen | Helpfulness of app for mentor | 4 |
| | Additional Training Suggestions | 5 |
| | Usefulness of training | 4 |

**Table 7. Percent change in mentees' anxiety, positive interpretation bias, and negative interpretation bias from pre-to-post intervention (per protocol sample).**

| Participant | Percent Change (Anxiety Symptoms) | Percent Change (Negative IB) | Percent Change (Positive IB) |
|---|---|---|---|
| P1 | - 33.3% | - 12.5% | + 15.8% |
| P5 | - 57.1% | - 3.7% | - 36.4% |
| P8 | - 14.3% | - 7.4% | + 9.1% |
| P11 | - 25% | 0 | - 16% |
| P12 | - 50% | 0 | - 21.7% |
| P14 | 0 | + 13.6% | + 15.8% |
| P15 | + 25% | + 5.3% | - 5.3% |
| P19 | - 25% | * | * |
| P21 | - 50% | * | * |
| P22 | - 33.3% | + 27.8% | + 12.5% |

*Note: \* indicates missing data.*

to 5). Neither benchmark for youth protocol adherence was met, with 40% of mentees (6/15) completing 50% of required sessions and only 30% of mentee (3/15) mentees completing 100% of required sessions.

Regarding *implementation framework feasibility* for mentors and youth, benchmarks were met for both measures of protocol adherence. At least 50% of weekly check-ins were completed by 60% of mentors (9/15), and 60% of mentors (9/15) discussed the MindTrails Teen app in at least 50% of completed check-ins.

**Acceptability (Hypotheses 1B, 2A, & 2B).** Our acceptability benchmarks were met or exceeded (e.g., ≥ 3; between "Neither agree or disagree" and "agree") for both mentors and youth on all ratings of *intervention acceptability* (mentees) and *implementation framework acceptability* (mentors and youth). Mean ratings can be found in Table 3.

**Intervention outcomes (Hypotheses 1C & 1D).** **Anxiety Symptoms:** The ITT sample comprised n = 15 participants, and the PP sample comprised n = 10 participants. Benchmarks were met for both the ITT and PP samples (PP: Hedge's $g$ = 1.16, 95% CI:[0.19, 2.13]; ITT: Hedge's $g$ = 0.88, 95% CI:[0.07, 1.69]). In both cases, effect sizes were large. The overall percent decrease from pre- to post-intervention in the PP and ITT samples was 28.3% and 26.2%, respectively. Percent change for each participant in the PP sample was calculated independently and is presented in Table 7 and Fig 1. Given literature suggesting that calculating Cronbach's alpha for two-item measures is not feasible, we opted to determine the split-half reliability of the GAD-2 by calculating the Spearman-Brown coefficient. Our results indicate an overall "good" reliability score of $r_2$ = 0.79 [67].

**Positive Interpretation Bias:** The ITT sample comprised n = 15 participants, and the PP sample comprised n = 8 participants. Benchmarks were met for both the ITT and PP sample (PP: Hedge's $g$ = 0.31, 95% CI:[-0.71, 1.33]; ITT: Hedge's $g$ = 0.60, 95% CI:[-0.26, 1.46]). The effect size for the PP sample was small and the effect size for the ITT sample was medium, suggesting an overall small-to-medium effect size for change in positive interpretation bias. The overall percent increase from pre-to-post intervention in the PP and ITT samples was 4.85% and 9.75%, respectively. Percent change for each participant in the PP sample was calculated independently and is presented in Table 7 and Fig 2. The results for the positive interpretation bias analyses should be considered in light of the poor consistency of the Recognition Rating Task.

**Negative Interpretation Bias:** The ITT sample comprised n = 15 participants, and the PP sample comprised n = 8 participants. Benchmarks were not met for either the ITT or the PP sample (PP: Hedge's $g$ = -0.12, 95% CI:[-1.14, 0.90]; ITT: Hedge's $g$ = 0.15, 95% CI:[-0.70, 1.00). The effect sizes for both samples were negligible, suggesting no

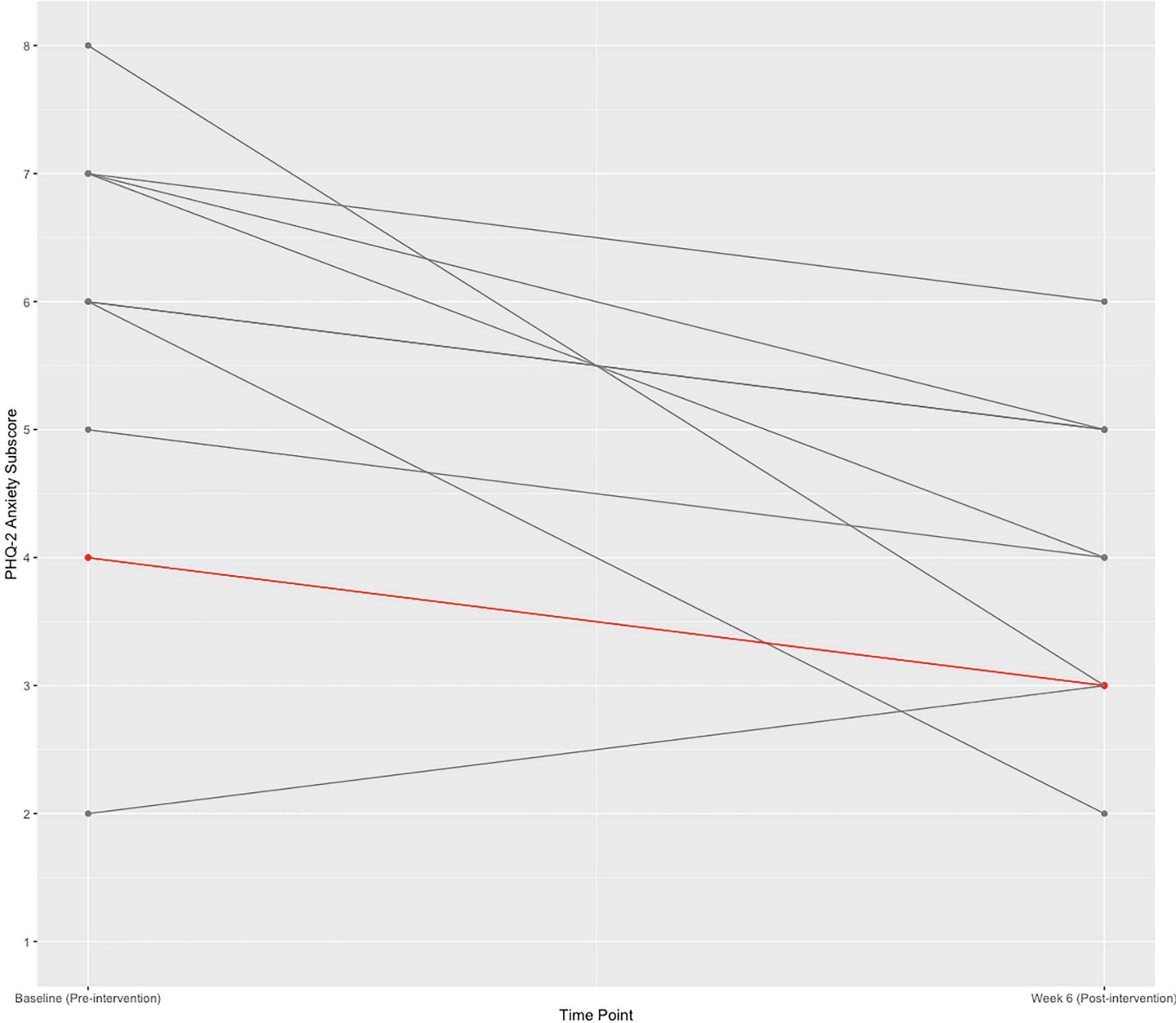

**Fig 1. Change in anxiety scores over time (per protocol sample).** Note: Grey lines indicate a single participant, and red lines indicate two participants who share the same change score. The scale of the y-axis ranges from 1–8 and scores represent the sum of the first two items on the PHQ-4 (i.e., the GAD-2).

reliable change in negative interpretation bias in either sample from pre-to-post intervention. Percent change for participants in the PP sample was calculated independently and is presented in Table 7 and Fig 3. The results for the negative interpretation bias analyses should be considered in light of the poor consistency of the Recognition Rating Task.

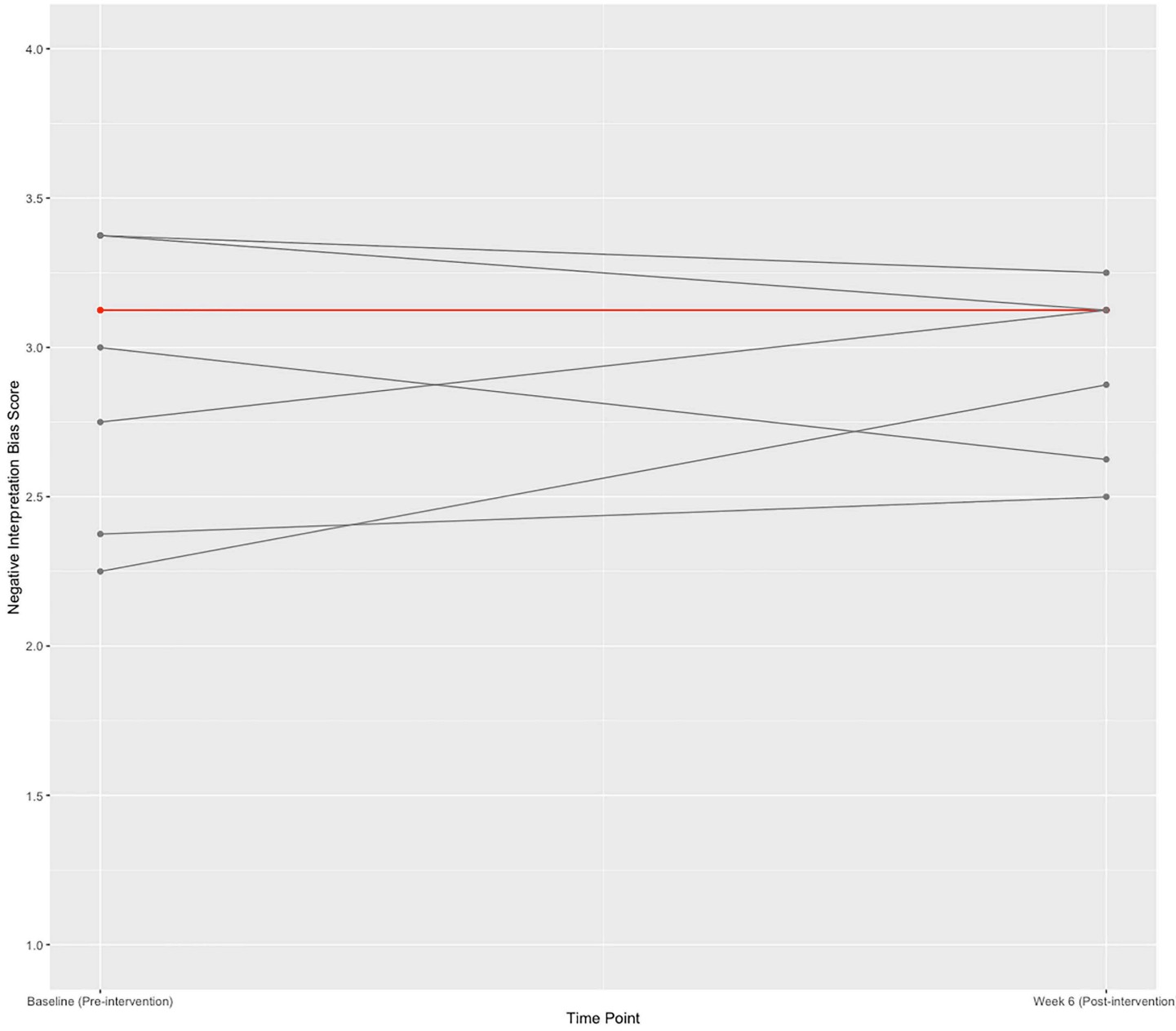

**Fig 2. Change in positive interpretation bias over time (per protocol sample).** Note: Grey lines indicate a single participant, and red lines indicate two participants who share the same change score The scale of the y-axis ranges from 1-4 and scores represent the average of all positive interpretation bias items on the Recognition Rating Task.

## Qualitative results

Qualitative results are presented in Tables 5 and 6. To protect youth privacy given the small study sample size, we have changed pronouns within all quotations to be gender neutral (e.g., [they/them]). See Supplementary Materials (S1 Text) for minor deviations from the qualitative preregistration.

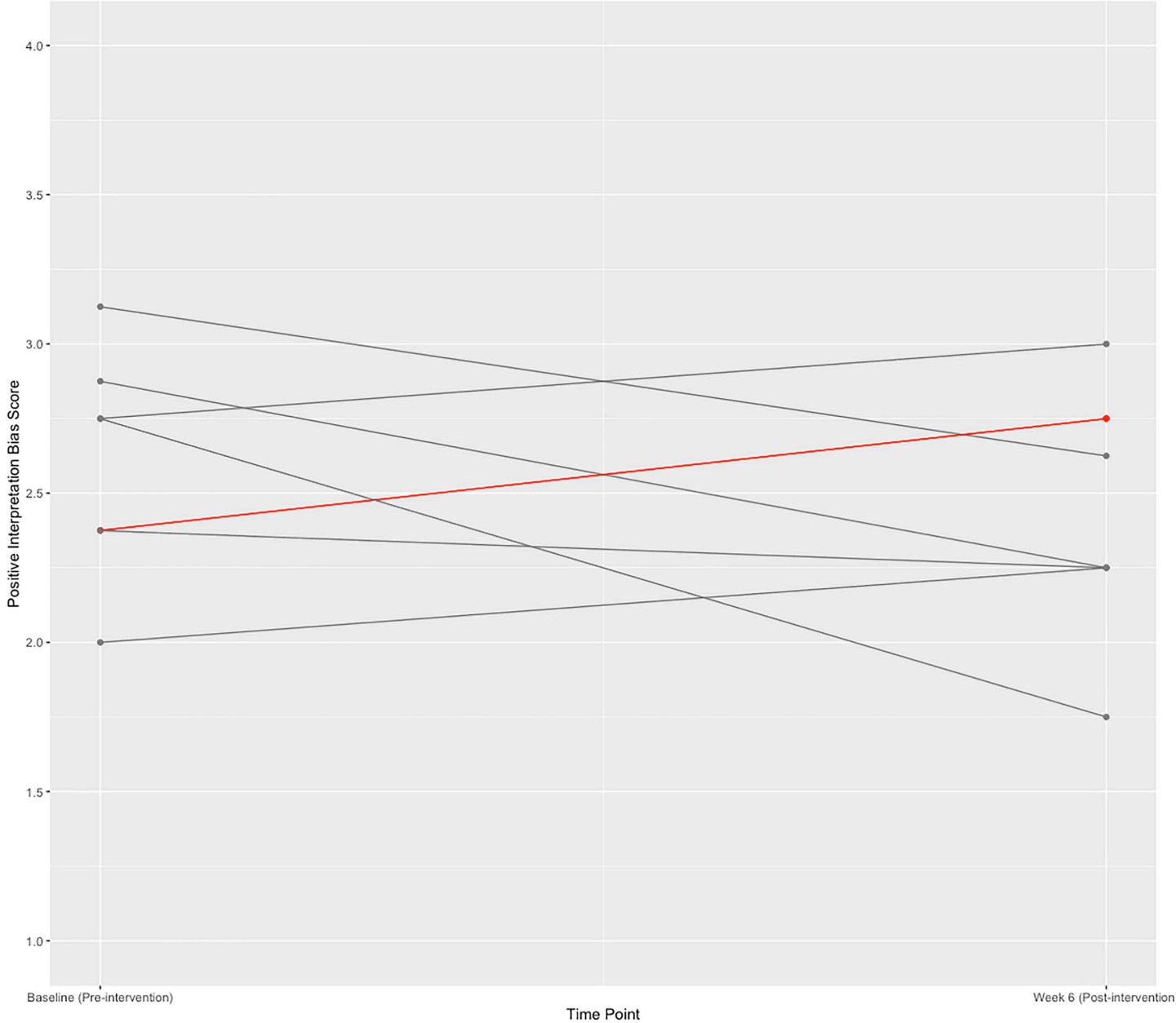

**Fig 3. Change in negative interpretation bias over time (per protocol sample).** Note: Grey lines indicate a single participant, and red lines indicate two participants who share the same change score The scale of the y-axis ranges from 1-4 and scores represent the average of all negative interpretation bias items on the Recognition Rating Task.

**Mentee themes.** **Theme 1: Facilitators of Engagement:** Among mentees who completed an interview, over half (4/7) discussed components of the app that facilitated their engagement, including a simple aesthetic design, clear instructions, and ease of use. Multiple users (3/7) specified the usefulness of push notifications in keeping them engaged:

*"… there were a lot of times the notification was very helpful because sometimes I would just forget about [the app], or it would be so busy…and then I'd just sit down and take the 15–20 minutes to do [the session]." (Mentee 1)*

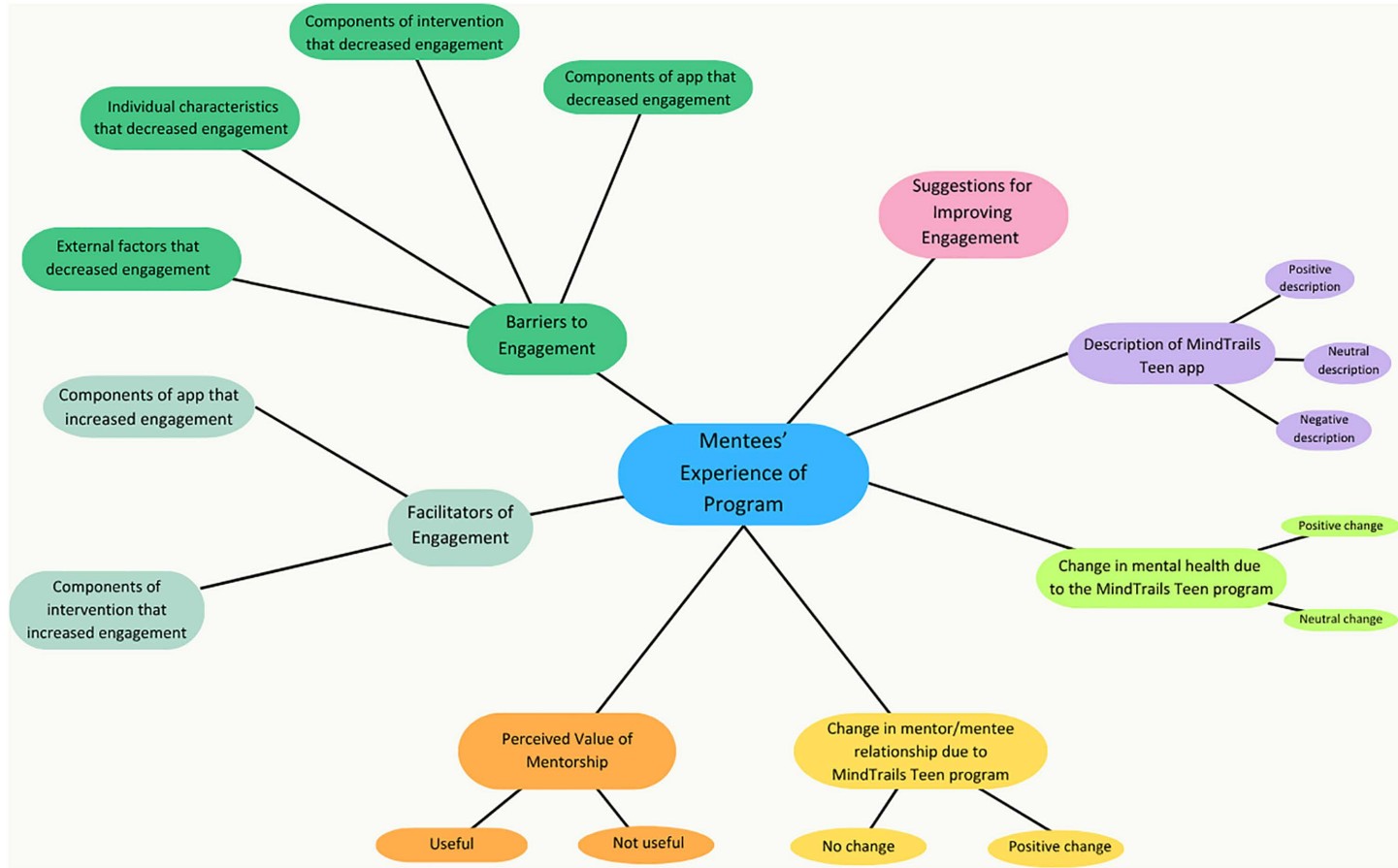

**Fig 4. Revised thematic map: Mentee version.**

Other mentees (5/7) discussed components of the intervention itself they found engaging, including the brevity of the intervention, the gamified components, and the relatability of the scenarios. Relatability of the CBM-I scenarios (4/7) was most often cited as an engaging factor of the intervention, with youth appreciating that the scenarios resembled anxiety-inducing situations in their daily lives:

> "*There was a technology one [that had] something to do with social media where your friends were going out without you, and they would post it and they didn't invite you and I liked that because stuff like that happens all the time…*" (Mentee 4).

**Theme 2: Barriers to Engagement:** Youth also identified specific barriers to engagement with the MindTrails Teen intervention. Some youth (3/7) identified barriers that were a function of the app itself, including technical glitches, an inability to return to the previous page without starting over, and the amount of storage required by the app. Other youth (6/7) were hindered by components of the intervention, including timers on certain sessions, the length and repetitiveness of the sessions, and the perceived unrelatability of the scenarios. The timers were brought up specifically by several youth (3/7). These were included in the "long" training sessions in which mentees were asked to spend sixty seconds writing out their thoughts and feelings in response to an ambiguous situation. The timer was initially included to encourage youth to stay on the page and not skip through the session, however, some youth expressed frustration with the timer:

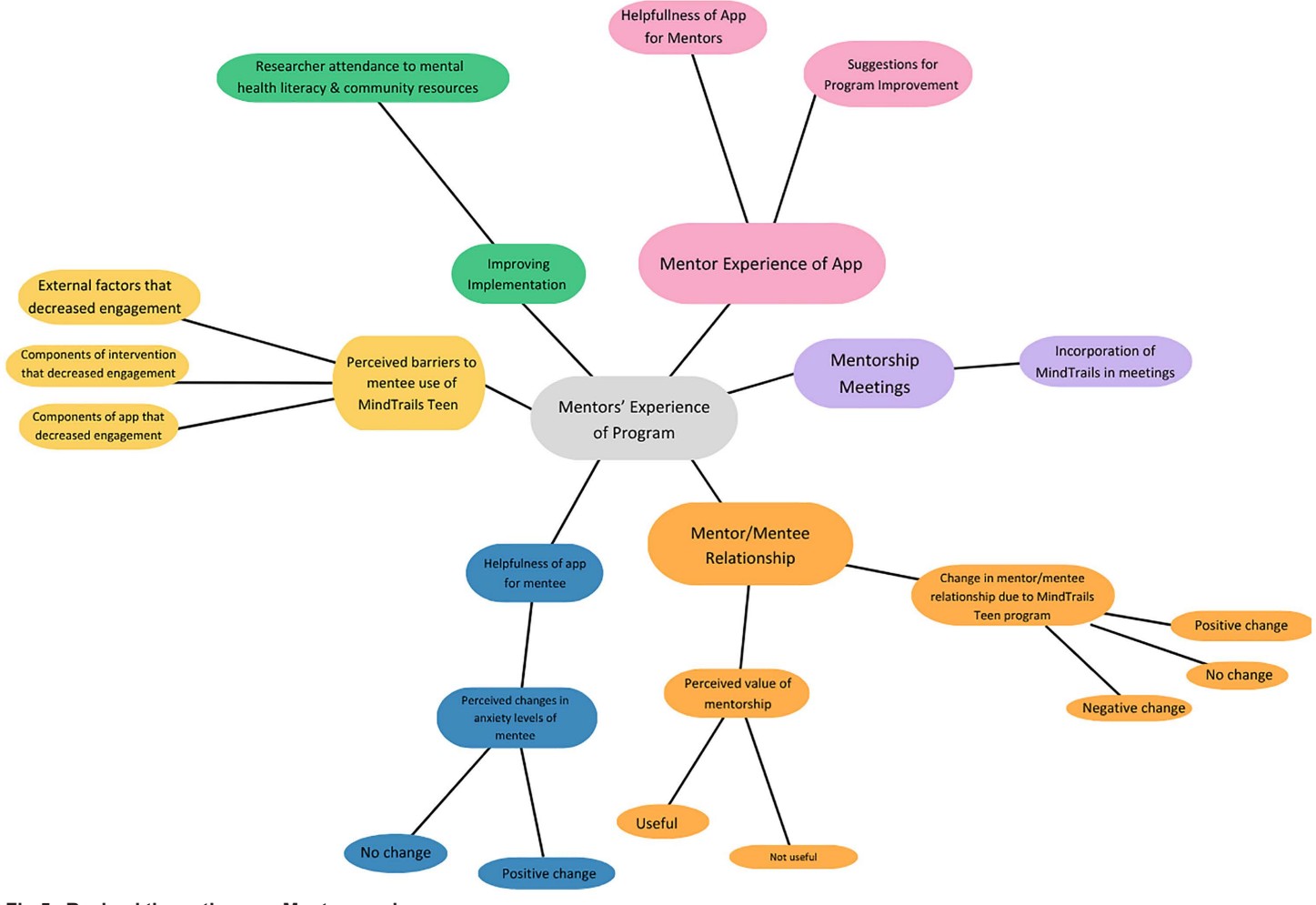

**Fig 5. Revised thematic map: Mentor version.**

*"I didn't really like the waiting times [after typing] because sometimes I can't really type more than one or two sentences, so I type something short and concise that conveys the meaning and then I was just sitting there waiting. It was a little bit tedious. (Mentee 4).*

Several youth (3/7) found the CBM-I scenarios monotonous and expressed a desire for more variation in content. Other youth (4/7) reported individual characteristics that served as barriers to engagement, including challenges with attention span, memory capacity, being too busy, or not having a private environment to complete the assigned session. Over half of youth (4/7) specifically identified short attention spans as a barrier to engagement.

**Theme 3: Suggestions for Improving Engagement:** All seven youth offered specific suggestions for improving engagement with the app. These included additional levels of content personalization, the ability to set a specific time to complete sessions, more variety in content, shorter sessions, additional gamification, improved accessibility features, and ability to customize colors and other app features. Youth overwhelmingly (7/7) preferred the short sessions (10–15 minutes) over the long sessions (15–20 minutes) and offered suggestions for optimal length of sessions, ranging from 5–10 minutes *(Mentee 2)* to 8–11 minutes in length *(Mentee 4).* Suggestions were also offered for app appearance and aesthetics. A brighter color

scheme was favored, and changes to the delivery of notifications (e.g., making the notification remain on the home screen vs. disappearing until the session was completed) were also proposed. In general, youth expressed a desire for as much customization as possible, both in terms of the CBM-I scenario content and in terms of app aesthetics and usage:

*"…being able to fully customize the app, place what buttons you want where, even having widgets that come pre-installed with the app, so that way you can click on the widget and instantly get to your next survey. There could be color changing aspects, there's a lot of things you could do with [the app]." (Mentee 4).*

Although several youth cited the relatability of the CBM-I scenario content as an engaging factor, others (3/7) desired a broader range of scenario content. Two youth reported a desire for increased gamification in the app:

*"Although not feeling anxious is a reward itself I feel like maybe some tokens that lead up to something as a reward could be nice." (Mentee 6)*

Three youth also suggested inclusion of content related to identity stress, including gender identity. However, one mentee also cautioned against including such stressors as scenarios, on the basis that they might do more harm than good if not properly implemented:

*"I would say be careful as those are heavy topics…swing the wrong way and they could hurt you more than help." (Mentee 3)*

**Theme 4: Change in mental health due to MindTrails Teen:** Over half of youth (4/7) discussed the impact of the MindTrails Teen app on their anxiety. Of these, all reported the app having a positive impact on their anxiety symptoms, and no youth reported iatrogenic effects. One youth reported that the timing of the app was helpful given their anxiety severity at the start:

*"[The app] was extremely helpful, especially since my anxiety just like hit its peak point right as I started the app, and then it started to go downhill and I now have a way to control it." (Mentee 1).*

When asked how they might describe MindTrails Teen to a friend, youth were equally split between positive (4/7) and neutral or mixed descriptions (4/7), with two youth providing both positive and neutral descriptions.

*"[MindTrails Teen] is an app that lets you talk about your anxiety and tries to get you to*

*be more comfortable with yourself and your surroundings."*

Among those who described the app more neutrally, youth tended to acknowledge its impact while citing its perceived weakness:

*"I would describe the app as long, kind of annoying, but impactful." (Mentee 4).*

**Theme 6: Mentor/Mentee Relationship:** Of those interviewed, most youth (6/7) discussed the impact of their mentor on their experience in the program. Over half of youth (5/7) cited their mentor as a positive influence and noted their function of keeping them engaged:

*"[My mentor] was on top of it and made sure I was doing well. He was a huge help with reminding me and knowing that someone was looking out [for me] was amazing." (Mentee 3).*

Another youth noted that it was especially helpful working with a mentor who also had anxiety:

"*Because she also has anxiety too, and she's tried ways to help me with it and then she just sent me this study one day and…it was just easy for me to get it done with someone I knew had gone through it.*"(Mentee 1)

Only one youth did not find working with a mentor as useful, citing a lack of content specifically related to the mentor or their relationship.

**Theme 7: Change in Mentor/Mentee Relationship due to MindTrails Teen program:** Over half of youth (5/7) discussed changes in their relationship with their mentor due to participation in the MindTrails Teen program. Slightly less than half of youth (3/7) endorsed positive changes in the relationship with their mentor while participating in MindTrails Teen:

"*We were already close, so I would say it just gave us more chances to bond and get to know each other better in all aspects.*" (Mentee 3).

Meanwhile, other youth (2/7) reported no changes in their relationship with their mentors. No youth reported a negative change in their relationship with the mentor as a function of participating in MindTrails Teen.

**Mentor themes. Theme 1: Perceived Value of Mentorship:** Over half of mentors (4/7) discussed their perceived usefulness to their mentee during the MindTrails Teen program. Of these four, three thought their presence was useful, and two specifically commented on how they played a different supportive role than a parent or family member would have:

"*I think having that sort of third person, that's not your dad, not your brother, not your sister, just any sort of third party to sort of bounce stuff off of is good…it's somebody that doesn't really have an investment in you – and, I mean, I have an investment in [the mentee] – but it's a sort of guiding relationship opposed to [their] dad needing [them] to do something. You never have that same level of stress.*" (Mentor 2).

Of the four who discussed this topic, only one mentor did not find their presence useful:

"*As far as I'm concerned, it was easy [for me] because it was so minimal…there wasn't much for me to offer.*" (Mentor 6).

**Theme 2: Change in Mentor/Mentee Relationship due to MindTrails Teen program:** Over half of mentors (5/7) discussed how their relationship with their mentee did or did not change as a function of participating in the MindTrails Teen program, with almost all (4/5) endorsing at least one positive relationship change. Two mentors discussed how participating in MindTrails Teen improved communication with their mentees about mental health. One of the two mentors also mentioned that the program was a helpful bridge to connect with her mentee over her own experiences of anxiety:

"*It felt good that I could tell [them] I have anxiety too…I liked that I could tell [them] that and [they] can know that adults go through this too and [they are] not the only one going through this.*" (Mentor 7)

Meanwhile, a minority of mentors (2/7) did not perceive a change in their relationship with their mentee as a function of participation. Only one mentor reported a negative change in their relationship with their mentee, due to their mentee being untruthful about their level of program completion.

"*It's the first time I [had] ever heard that [they] lied to me so that's not good. But, to look at the bright side, it's going to give me an opportunity to talk to [them] about that so [they] don't do it again.*" (Mentor 5)

**Theme 3: Mentorship Meetings:** Almost all mentors (6/7) discussed their experience of and opinions on meeting with their mentees during the MindTrails Teen program. Of these, all six reported brief discussion of MindTrails Teen in weekly meetings and/or check-ins but had diverse reasons for doing so. Three mentors cited the check-ins as a method of offering support and keeping their mentees accountable. Meanwhile, one mentor reported keeping the check-ins short so that they did not interfere with their existing relationship with their mentee, and another mentor cited doing only a brief content discussion given a reticent mentee.

**Theme 4: Benefit of App to Mentee Mental Health:** Over half of mentors (5/7) discussed the perceived impact of the MindTrails Teen app on their mentees' mental health. Of these five, four reported a positive impact:

*"From my understanding it actually did help [them] overcome a few things like some of [their] anxieties because [the mentee] did say [they were] able to use some of it in [their] daily life. (Mentor 6)*

No mentors identified negative effects of the MindTrails Teen program on mentee mental health, and only one mentor suggested the app as having a neutral or no impact.

**Theme 5: Barriers to Mentee Engagement:** Over half of mentors (5/7) discussed perceived barriers to their mentees' engagement in MindTrails Teen. All five noted barriers related to external influences, including the timing of the intervention (e.g., holidays, summer vs. school year), boredom of mentee, pressure from parents to participate, and competing demands within teens' lives:

*"I think that the 5 weeks was difficult because it's hard to get in a routine with something in just 5 weeks. Plus, you have to add in, we [had] Christmas, we had holidays in there, which was probably a bad time to start." (Mentor 3).*

Two mentors noted individual characteristics of their mentees that may have impacted engagement, including stressors not appropriate for the app and personality factors of their mentees (e.g., people pleasing tendencies). Finally, three mentors indicated components of the app or intervention that hindered their mentees engagement (e.g., mentees saying they found the app repetitive).

**Theme 6: Suggestions for Program Improvement:** Almost all mentors (6/7) provided suggestions for program improvement. These included giving mentors access to the app, having more specific prompts to engage mentees during meetings, providing a more structured guide to the app and content, sending notifications to mentors when a mentee completed a session, expanding the age range of the app, and attending to local implementation barriers (e.g., level of mental health literacy in the community). Giving the app to mentors was a common suggestion for improvement, with over half of mentors (4/7) stating that this would have benefited them.

*"I wish I could've seen what [they were] doing because then that would've been really helpful…I was like, 'so how's it going', and [they were] like, 'yea it's good, like I did them all,' but I couldn't prod any further." (Mentor 1)*

Beyond just having the app, some mentors (3/7) thought that being able to see mentee progress on a dashboard would have made them feel more involved, and one mentor suggested that specific prompts of what to talk about each week would have been beneficial. Finally, two mentors (2/7) mentioned that mental health literacy and access to mental health resources generally in the local community was something researchers should consider when implementing the study, particularly in rural areas.

**Theme 7: Mentor Experience of the MindTrails Teen Program:** Over half of mentors (5/7) discussed their personal experience of participating in the MindTrails Teen program. Of these five, four discussed how participation benefited them as mentors. Two mentors appreciated how the app provided them an additional tool to support their mentees:

*"I think a lot of times as mentors we can struggle with: how do I help? And this gives you a tool. This gives you something to focus on and then not [have it be] about me lecturing [them]. More of a 'hey let's pull this up and do this together' or 'how about you go sit down and do this for a few minutes and then we can talk again.' It just felt like a good tool in my toolbox, I guess." (Mentor 5).*

Over half of mentors (4/7) also discussed their experience of the pre-program mentor training, with all four stating that they found it useful. One mentor specifically stated that she liked being trained in a group of other mentors:

*"I think [the training] was super informative and I don't know if you usually have sessions with multiple people or if its more one on one but I do like…that there were two additional [mentors] on the same call with me." (Mentor 4)*

Mentors (5/7) also provided specific suggestions to improve training in the future, including creating a more comprehensive packet of training resources (e.g., adding specific prompts to encourage conversation in check-in meetings), and offering mentors in the study the opportunity to meet and talk to one another.

## Discussion

The present study investigated the feasibility and acceptability of MindTrails Teen, a digital cognitive bias modification for interpretation (CBM-I) intervention implemented via a 5-week, mentor-guided pilot open trial. To our knowledge, this is the first test of a mentor-supported DMHI for teens with anxiety symptoms. Quantitative pre-post analyses indicate moderate improvements in anxiety, small improvements in positive interpretation bias, and no change in negative interpretation bias. These findings should be interpreted with caution given the poor internal consistency of the interpretation bias measure used and the absence of a control group. The study met all benchmarks for implementation framework feasibility, but only one out of two benchmarks for intervention feasibility (benchmarks were met for clinical deterioration but not protocol adherence). All acceptability benchmarks related to the intervention and implementation framework were met for both mentors and mentees.

Qualitative results, considered in light of demand characteristic and social desirability bias, support a possible anxiolytic effect of the intervention and include suggestions for app improvement (e.g., making sessions shorter, increasing personalization of app features and training content, and including more variety of content). Youth noted both facilitators to their engagement, including app notifications and relatability of content, as well as barriers, including timers on exercises, repetitiveness of content, and self-reported poor attention span. Mentors reported positive changes to their relationship with their mentees while using MindTrails Teen, usefulness of the app to facilitate mentoring, implementation barriers, and suggestions for program improvement (e.g., giving mentors access to the app).

### Intervention feasibility and acceptability

Clinical deterioration benchmarks were met for mentees, suggesting that MindTrails Teen is safe and unlikely to lead to iatrogenic effects. Protocol adherence benchmarks were met for mentors but not for mentees. Post-hoc descriptive analyses indicate the mean number of sessions completed by mentees in the trial was 4.7, and the median number of completed sessions was 3, suggesting that most participants received at least a third of the intended dose of 10 sessions. This degree of protocol adherence was less than expected and is clearly a target for improvement for future versions of the app and future trials, but is actually not uncommon in other trials examining the feasibility of DMHIs for youth in real-world implementation settings [68]. Increasingly, research indicates that individuals with differing levels of adherence and engagement can benefit from DMHIs [69] and there is not a "one size fits all" approach to use. Still, it is worthwhile to interrogate our low adherence rates and explore the function of

mentors promoting sustained adherence as well as initial engagement. Although our trial included adherence-boosting measures such as compensation and human support, our sample had several characteristics that may have impacted engagement. First, participants were selected only for elevated anxiety symptoms, and community samples occasionally evidence less robust treatment adherence rates compared to those recruited from schools or mental health clinics. Relatedly, our sample reported mild-to-moderate anxiety on average ($M = 9.32$ on the GAD-7 from the eligibly survey), which is higher on average than youth users of DMHIs recruited from unselected samples [70]. Although some studies associate symptom severity with greater adherence [69], others have identified a curious relationship between symptom improvement and early drop-off, positing that early gains in a DMHI may unintentionally discourage continued use [71,72]. This may be especially true for our intervention, in which repetitive content was frequently cited as a barrier for youth engagement—perhaps once youth felt they understood and benefited from the paradigm, they no longer felt motivated to engage. Although previous single-session CBM-I interventions have not proven effective [23], the steep drop-off in engagement in our intervention highlights the challenge of determining the optimal number of sessions. Finally, mentors may have varied in their motivation or willingness to engage in structured activities.

Mentor feedback suggesting additional information and resources may be helpful for future intervention development. Allowing mentors access to the intervention content and providing them with just-in-time notifications of training completion could have allowed them to provide more timely support and encouragement, which may have boosted participants' training completion. In fact, research suggests that more immediate rewards, compared to delayed rewards, are linked to greater intrinsic motivation [73]. For example, an immediate virtual high-five from a mentor for completing a training session may help engagement. Regardless, this unmet benchmark suggests there were barriers to youth adherence that were not fully ameliorated by our added mentor human support.

Interviews with participating youth indicate they found the app acceptable and beneficial to their anxiety symptoms. When asked if they would recommend the app to other teens, the average youth response fell between "Agree" and "Strongly Agree". Youth were less positive, but still within the acceptable range, when asked if they enjoyed using the app and if they thought the app changed their thinking in anxiety-inducing situations, with average responses for both items falling between "Neither agree nor disagree" and "Agree". Interestingly, despite less enthusiastic endorsement of the app for their own use, youth were still highly likely to recommend the app to others struggling with anxiety, perhaps implying perceived benefit, if not equivalent personal enjoyment. Given we only interviewed a subset of the sample, youth responses to the interviewer should also be considered in light of extroversion and social desirability bias.

## Implementation framework feasibility and acceptability

Among those who completed the study, the implementation framework (i.e., youth app use guided by a mentor) was both feasible and acceptable. Mentors rated their enjoyment of the app and their willingness to recommend the app to other mentors more highly than they rated their own preparedness to support their mentee during the program. This is corroborated by qualitative feedback suggesting mentors enjoyed the program but could benefit from additional supports by the research team (e.g., discussion prompts for meetings with youth, opportunities to connect with other mentors). Youth also rated the implementation framework highly, particularly enjoying working with their mentor and finding their mentor helpful during the program.

## Intervention outcomes

Unexpectedly, changes in positive and negative interpretation bias (small effect size and no change, respectively) were smaller than change in anxiety symptoms. Results from previous trials of digital CBM-I for youth typically demonstrate the opposite pattern, in which change in anxiety is achieved less consistently than change in positive and negative interpretation bias [19,22,24]. Given the relative consistency of interpretation bias change observed in prior trials of digital CBM-I,

we suspect that the poor internal consistency of our novel measure of interpretation bias is likely to explain our mixed findings for bias change. We adapted the Recognition Rating Task (our measure of interpretation bias, which was originally designed for adults) so the scenario content was more suitable for adolescents. Our post-hoc discovery of the measures' poor internal consistency highlights the importance of selecting psychometrically valid instruments for pilot or proof-of-concept studies. While it remains possible that our low protocol adherence reduced target engagement, this seems unlikely given the limited evidence that number of CBM-I training trials predicts outcomes [74].

Of course, the open trial design means we cannot confidently conclude that the CBM-I training caused reductions in anxiety. Besides time or regression to the mean, increasing interactions with their mentor or simply enrolling in a study on anxiety may have contributed to youth improvement on the GAD-2. It is also possible that change in the mentoring relationship during the study may also have contributed to the anxiety reduction. While not a primary outcome, we did collect data on mentor/mentee relationship quality at each time point (see Supplementary Materials; S1 Text). Exploratory post-hoc analyses indicate a small to medium improvement in strength of that relationship in both the PP and ITT samples. While not itself evidence of mechanistic effects, presence of a supportive mentor is linked to improvements in youth anxiety [75,76], raising the possibility that relationship improvement (plausibly brought on by the app, based on the qualitative feedback) was a mechanism that contributed to the anxiety reduction. Along these lines, qualitative findings highlighted that both mentors and youth reported positive benefit from discussing shared symptoms of anxiety. Reciprocal self-disclosure in mentoring relationship—when used judiciously—can be a way for mentors to demonstrate trust and closeness with their mentees [77]. Furthermore, talking about mental health with trusted confidantes can reduce symptoms and stigma among youth [78–80]. While these mechanisms require further testing in a larger controlled trial, they offer preliminary support for mentor-supported implementation of DMHIs.

## Limitations

Although the study had multiple strengths, including its mixed methods design, recruitment of an anxious sample, and innovative implementation framework, there were also several limitations. First, our sample of mentors was racially and ethnically homogenous (primarily White and non-Hispanic/Latino) and though our sample of mentees had slightly more racial and ethnic diversity than the mentors, it was still not fully representative of the national mentee population. Furthermore, while our sample had unique representation from underserved rural communities (Iowa, Nebraska, South Texas) it did not contain broad geographic diversity. Second, our small sample size significantly limits the generalizability of our findings. More research is needed with large, geographically and demographically diverse samples randomized to MindTrails Teen vs. a comparison condition before making conclusive recommendations about the efficacy of the intervention and the feasibility of the implementation pathway. Third, it is possible that social desirability bias and demand effects may have accounted for some of our findings, with teens feeling some pressure to report positive changes in symptoms and their mentor-mentee relationship following the program, especially given the pre-established relationships between mentors and mentees. Future work should measure the influence of social desirability on the intervention effects, and take additional steps to reduce demand effects, such as having an independent assessor. Fourth, the lack of a control/comparison group, while a reasonable choice for this preliminary pilot evaluation, is a significant limitation. Fifth, as noted, measurement issues, especially the low reliability of the interpretation bias measure, make it more difficult to validly test effects.

Finally, recruitment difficulties raise questions about implementation framework feasibility. Alongside targeted social media campaigns, our team contacted 140 individual organizations, only two of which led to significant enrollment. Moving from recruitment to enrollment was also difficult given the need to consent parents, youth, and mentors. Out of the 41 parent/mentee/mentor triads sent at least one consent form, 14 of these triads only made it partially through the consent process (i.e., at least one of the 3 members did not sign, preventing enrollment

from proceeding). This may be in part because some models of mentoring prioritize the relationship as the goal, as opposed to the relationship serving as context for goal setting and achievement. Although recent research shows the benefits of goal-setting and targeted mentoring [37,38,81], there is some resistance to mentoring programs leveraging the relationship to achieve goals [34]. Mental health-focused interventions may be seen as outside of the scope of traditional mentoring relationships, with mentors potentially gravitating toward recreational activities that present lower perceived risk and are more appealing to mentees. It may be the case that formal youth mentoring programs that prioritize the relationship may not be the appropriate outlet for implementing our intervention. Therapeutic mentoring [82] – paraprofessional mentors serving as a trained and paid support for youth in treatment – may offer a more suitable framework for future implementation. In addition to exploring this avenue, future work should prioritize forging partnerships with mentoring organizations prior to study recruitment and working directly with key implementation stakeholders (e.g., program directors, mentors) to evaluate and improve the proposed recruitment and implementation strategy. Our difficulty recruiting mentor/mentee dyads from individual programs may suggest that top-down recruitment approaches (e.g., partnering with parent organizations) will be more fruitful than recruiting from small, local branches. It will be critical to establish reliable recruitment pathways for MindTrails Teen for it to one day serve as a feasible, accessible, and scalable intervention.

## Future directions

Based on qualitative feedback, future versions of the app should prioritize shorter sessions, additional content personalization, and increased gamification. Artificial intelligence (AI) may be a logical future avenue for improving personalization (e.g., tailoring scenario content to youth demographic information and domains of concern, assuming this can be done ethically and effectively preserve privacy). Although the mentor delivery model was feasible and acceptable to those users who enrolled and remained engaged, future work should seek to streamline the consent process to be less burdensome for participants and engage with mentoring program directors and other key stakeholders to understand organization-level barriers to participation.

Mentors referenced several important implementation considerations, such as the possibility that parents pressure youth to participate in mental health programs and the limited mental health literacy in the community. Most mentor/mentee dyads hailed from areas of the US less commonly represented in clinical research (Iowa, South Texas, and Nebraska) [83]. Though we lack data on whether participants identified as urban or rural, the response rate from less commonly represented regions suggests the app may help address an unfilled need for youth mental health services in these communities.

Another traditionally-underserved user group that was well represented in the current study are gender non-conforming youth, who made up nearly one-third of our very small sample. We did not specifically outreach to this population when recruiting for this study, and only selected for youth with elevated symptoms of anxiety. This finding dovetails with the elevated rates of anxiety among trans and gender non-conforming youth [84], and suggests a critical need for support for this community. While CBM-I is not necessarily an ideal tool in its current form to directly address anxiety stemming from stigma and discrimination (because of concerns that attempts to shift users' thinking in these contexts could come across as gaslighting about legitimate threats), future versions of the app should provide additional resources for both the youth facing stressors related to their identities and to their mentors.

Finally, in addition to refining the mentorship supportive accountability model, future research should explore other implementation pathways and potential options for scaling the intervention. While the mentorship implementation framework holds promise for increasing supportive accountability adherence and may increase access by providing a point of care in spaces where teens already interact, scaling of the mentorship implementation framework is inherently limited by mentor availability and youth having access to participating mentorship programs. School mental health counselors may be able to fulfil a comparable role to mentors in a school-based implementation framework, and use of the app in peer-support or collegiate

mentoring programs could capitalize on the reported usefulness of disclosing and discussing shared lived experiences of mental health. These and other approaches could be investigated for feasibility, acceptability, and efficacy.

## Conclusion

Youth participating in the MindTrails Teen program with mentor guidance reported decreases in anxiety from pre-to-post intervention. Invalidity of our CBM-I measure prevented us from drawing conclusions about pre-to-post change in CBM-I. The app and intervention delivery model were overall feasible and acceptable to youth and mentors based on predetermined benchmarks, though youth failed to meet the benchmark for protocol adherence. Qualitative results corroborate the quantitative findings, reiterating the positive impact of participation for both mentors and teens and offering concrete suggestions for future changes to the app and implementation approach. Future research should test the impact of MindTrails Teen in different environments, with and without different types of human support, and in a larger randomized controlled trial seeking to clarify the impact of the program on target engagement.

## Supporting information

**S1 Text. 1: Methods Notes:** 1: Additional Recruitment Information, 2: Mentor Training Details, 3: Additional CBM-I Training Tasks. **2: Results Notes:** 1: Mentor/Mentee Relationship Quality, 2: Deviations from Pre-registration. **3: Interview Guides:** 1: Mentee Interview Guide, 2: Mentor Interview Guide. **4: Supplementary Tables:** S1 Table A: Data Triangulation Table, S1 Table B: Measure Administration Schedule. **5: Supplementary Figures:** S1 Fig A: Initial Thematic Map: Mentee Version, S1 Fig B: Initial Thematic Map: Mentor Version.
(DOCX)

## Author contributions

**Conceptualization:** Alexandra Werntz, Jean E. Rhodes, Bethany A. Teachman.

**Data curation:** Emma Wolfe, Audrey Michel, Yiyang Zhang, Mark Rucker, Mehdi Boukhechba, Laura E. Barnes, Bethany A. Teachman.

**Formal analysis:** Emma Wolfe.

**Funding acquisition:** Alexandra Werntz, Laura E. Barnes, Jean E. Rhodes, Bethany A. Teachman.

**Investigation:** Emma Wolfe, Alexandra Werntz, Bethany A. Teachman.

**Methodology:** Emma Wolfe, Alexandra Werntz, Mark Rucker, Mehdi Boukhechba, Bethany A. Teachman.

**Project administration:** Emma Wolfe.

**Resources:** Mark Rucker, Mehdi Boukhechba, Laura E. Barnes, Bethany A. Teachman.

**Software:** Mark Rucker, Mehdi Boukhechba.

**Supervision:** Alexandra Werntz, Bethany A. Teachman.

**Visualization:** Emma Wolfe.

**Writing – original draft:** Emma Wolfe.

**Writing – review & editing:** Emma Wolfe, Alexandra Werntz, Jean E. Rhodes, Bethany A. Teachman.

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
