## [Decision Letter · Decision Letter 0]

18 May 2025

Response to Reviewers
Revised Manuscript with Track Changes
Manuscript
**Journal Requirements:**
**Additional Editor Comments (if provided):**
**Reviewers' Comments:**

**Comments to the Author**

1. Does this manuscript meet PLOS Digital Health’s publication criteria?

Reviewer #1: Yes

Reviewer #2: Yes

2. Has the statistical analysis been performed appropriately and rigorously?

Reviewer #1: Yes

Reviewer #2: Yes

3. Have the authors made all data underlying the findings in their manuscript fully available (please refer to the Data Availability Statement at the start of the manuscript PDF file)?

Reviewer #1: Yes

Reviewer #2: Yes

4. Is the manuscript presented in an intelligible fashion and written in standard English?

Reviewer #1: Yes

Reviewer #2: Yes

Reviewer #1: I recommend clarifying the rationale behind selecting the 13 to 17 age group. According to the United Nations, there is no universally agreed-upon definition of "youth." However, for statistical purposes, the UN defines youth as individuals aged 15 to 24 years. It would be helpful to explain how your chosen age range aligns with or diverges from this definition.

Additionally, please ensure that data is included in rows 217–220 of the table.

Reviewer #2: This study provides a great example on how mentor guided digital mental health intervention can help young people thrive.

The sampling approach uses a convenience sampling resulting to a very low sample size, explain any strategy or strategies put in place to enhance representativeness, reduce bias and broaden the reach.

In line 262, it was mentioned that all participants completed the demographic questionnaire, this contradicts what was presented earlier in the study as well as in table 1.

Could you clarify the specific stage at which the reduction in anxiety symptoms was measured? was this outcome assessed at the conclusion of the intervention or during a subsequent follow-up period?

**Do you want your identity to be public for this peer review?** For information about this choice, including consent withdrawal, please see our Privacy Policy

Reviewer #1: No

Reviewer #2: No

**Figure resubmission:****Reproducibility:** To enhance the reproducibility of your results, we recommend that authors of applicable studies deposit laboratory protocols in protocols.io, where a protocol can be assigned its own identifier (DOI) such that it can be cited independently in the future. Additionally, PLOS ONE offers an option to publish peer-reviewed clinical study protocols. Read more information on sharing protocols at https://plos.org/protocols?utm_medium=editorial-email&utm_source=authorletters&utm_campaign=protocols

---

## [Decision Letter · Decision Letter 1]

19 Aug 2025

Response to Reviewers
Revised Manuscript with Track Changes
Manuscript
**Journal Requirements:**
**Additional Editor Comments (if provided):**
**Reviewers' Comments:**

**Comments to the Author**

Reviewer #1: All comments have been addressed

Reviewer #3: All comments have been addressed

Reviewer #4: (No Response)

Reviewer #5: All comments have been addressed

publication criteria?

Reviewer #1: Yes

Reviewer #3: Yes

Reviewer #4: No

Reviewer #5: Yes

3. Has the statistical analysis been performed appropriately and rigorously?

Reviewer #1: Yes

Reviewer #3: Yes

Reviewer #4: Yes

Reviewer #5: Yes

4. Have the authors made all data underlying the findings in their manuscript fully available (please refer to the Data Availability Statement at the start of the manuscript PDF file)?

Reviewer #1: Yes

Reviewer #3: Yes

Reviewer #4: Yes

Reviewer #5: Yes

5. Is the manuscript presented in an intelligible fashion and written in standard English?

Reviewer #1: Yes

Reviewer #3: Yes

Reviewer #4: Yes

Reviewer #5: Yes

Reviewer #1: (No Response)

Reviewer #3: 1. The small sample size is acknowledged as a pilot limitation, but more discussion is needed on how this affects generalizability, especially given the racial homogeneity and limited geographical spread of participants.

2. The rationale for using mentor-guided delivery of a DMHI is compelling and timely; however, clarify whether mentors had any prior training or experience in delivering psychological interventions beyond the brief training provided.

3. Consider expanding on why negative interpretation bias did not change—was this likely due to the psychometric flaws in the task or limitations in the intervention content?

4. Table 2 is well-structured, but there are readability issues in formatting. Consider separating qualitative and quantitative benchmarks to improve clarity.

5. The selection of the 13–17 age group is now explained, but referencing specific developmental literature on adolescent cognitive-emotional development would strengthen the justification.

6. The manuscript would benefit from a short paragraph on implications for scalability—especially since mentor-based models may limit reach in real-world settings.

7. Include limitations related to potential social desirability bias in mentor and youth reporting, especially as dyads had established relationships prior to study participation.

8. Given the pilot design, consider framing the positive findings on anxiety reduction more cautiously—refrain from implying causality without a control group.

Reviewer #4: This manuscript describes a very small, single-group, pre/post pilot study of a cognitive bias app for anxiety among teens receiving mentorship. Overall, while the study could make a modest contribution to the literature, a few major concerns should be addressed.

1.) The weakness of the cognitive bias measure remains understated. Although the manuscript repeatedly says that the relevant findings "should be interpreted with caution," a reliability of a=0.35 suggests that the measure largely reflects error and likely cannot be interpreted at all. If reliability was really this low, then it would seem any findings (e.g., pre-post effect sizes) are mostly random. If the authors elect to retain these analyses, they should thoroughly and meaningfully address what any findings based on it mean, given that the measure is largely error.

2.) Reliabilities for other measures should be reported as well, especially the PHQ-9 anxiety questions.

3.) Similarly, given that the key anxiety outcome here was only two items from the PHQ-9, some justification should be provided about why this measure was chosen as an outcome relative to other, more well-validated and robust measures of anxiety.

4.) It's not entirely clear how this study is "mixed methods" research, other than simply using both qualitative and quantitative methods to evaluate the app. If the authors want to refer to the study as mixed methods, they should explain what approach they took (exploratory sequential, explanatory sequential) and how the results of the quantitative results informed the qualitative work or vice versa.

5.) Some of the conclusions overstate the strength of evidence these findings provide supporting the promise of the app. For example, the discussion says "Qualitative results corroborate the quantitative anxiolytic effect found for the app," suggesting that the data show that the app is responsible for the changes in anxiety. With most participants using the app only a few times (how long in duration, total?), why would app use be most likely to explain the changes in anxiety? Other potential explanations for this decrease in anxiety are not meaningfully discussed (other than briefly mentioned in the limitations), such as increasing the frequency of interactions with their mentor or even simply enrolling in a study on anxiety. As another example, referring to changes in anxiety as "robust" may also be an overstatement, given that the study only collected a two-item measure of anxiety at two timepoints among 10-15 participants.

6.) Finally, I realize that some of the difficulty recruiting participants may have been exclusively due to research-related procedures (e.g., the need to get consents from parents, mentors, and assent from mentees, the need for live training, etc.), but framing these results as though they support the feasibility of this approach in practice may also be a stretch, given that so few agreed to participate. The potential implications for this finding, and its implications for scaling this model outside the research context, should be meaningfully discussed as well.

Reviewer #5: The authors have done a thorough job addressing all previous reviews.

**Do you want your identity to be public for this peer review?** For information about this choice, including consent withdrawal, please see our Privacy Policy

Reviewer #1: No

Reviewer #3: **Yes:** Bala Nimmana

Reviewer #4: None

Reviewer #5: No

**Figure resubmission:****Reproducibility:** To enhance the reproducibility of your results, we recommend that authors of applicable studies deposit laboratory protocols in protocols.io, where a protocol can be assigned its own identifier (DOI) such that it can be cited independently in the future. Additionally, PLOS ONE offers an option to publish peer-reviewed clinical study protocols. Read more information on sharing protocols at https://plos.org/protocols?utm_medium=editorial-email&utm_source=authorletters&utm_campaign=protocols

---

## [Decision Letter · Decision Letter 2]

24 Nov 2025

Response to Reviewers
Revised Manuscript with Track Changes
Manuscript
**Journal Requirements:**
**Additional Editor Comments (if provided):**
**Reviewers' Comments:**

**Comments to the Author**

Reviewer #3: All comments have been addressed

Reviewer #6: All comments have been addressed

Reviewer #7: All comments have been addressed

publication criteria?

Reviewer #3: Yes

Reviewer #6: Yes

Reviewer #7: Yes

3. Has the statistical analysis been performed appropriately and rigorously?

Reviewer #3: Yes

Reviewer #6: Yes

Reviewer #7: Yes

4. Have the authors made all data underlying the findings in their manuscript fully available (please refer to the Data Availability Statement at the start of the manuscript PDF file)?

Reviewer #3: Yes

Reviewer #6: Yes

Reviewer #7: Yes

5. Is the manuscript presented in an intelligible fashion and written in standard English?

Reviewer #3: Yes

Reviewer #6: Yes

Reviewer #7: Yes

Reviewer #3: The revisions meaningfully strengthen the manuscript, especially the clearer limitations and softened causal language.

Your clarification regarding the poor reliability of the interpretation-bias measure is appreciated.

Separating the feasibility and acceptability tables improves readability

The mixed-methods integration is clearer, though one summarizing sentence in the main text could help readers quickly understand your triangulation approach.

Reviewer #6: The paper is a bit long, it could easily have been two papers, one quantitative and one qualitative, but I guess all comments have been addressed satisfactorily so this should be accepted now

Reviewer #7: The authors have provided a thorough and comprehensive response to the review comments and they have successfully addressed the major concerns regarding the limitations of the study.

I believe the manuscript is now significantly improved and meets the necessary scientific standards for publication.

**Do you want your identity to be public for this peer review?** For information about this choice, including consent withdrawal, please see our Privacy Policy

Reviewer #3: **Yes:** Bala Nimmana

Reviewer #6: No

Reviewer #7: No

**Figure resubmission:**

**Reproducibility:** To enhance the reproducibility of your results, we recommend that authors of applicable studies deposit laboratory protocols in protocols.io, where a protocol can be assigned its own identifier (DOI) such that it can be cited independently in the future. Additionally, PLOS ONE offers an option to publish peer-reviewed clinical study protocols. Read more information on sharing protocols at https://plos.org/protocols?utm_medium=editorial-email&utm_source=authorletters&utm_campaign=protocols

---

## [Decision Letter · Decision Letter 3]

22 Dec 2025

A Mixed Methods Evaluation of a Pilot Open Trial of a Mentor-Guided Digital Intervention for Youth Anxiety

PDIG-D-25-00190R3

Dear Ms. Wolfe,

We are pleased to inform you that your manuscript 'A Mixed Methods Evaluation of a Pilot Open Trial of a Mentor-Guided Digital Intervention for Youth Anxiety' has been provisionally accepted for publication in PLOS Digital Health.

Best regards,

Haleh Ayatollahi

Section Editor

PLOS Digital Health

**Additional Editor Comments (if provided):**

**Reviewer Comments (if any, and for reference):**

Reviewer's Responses to Questions

**Comments to the Author**

Reviewer #3: All comments have been addressed

Reviewer #6: All comments have been addressed

publication criteria?

Reviewer #3: Yes

Reviewer #6: Yes

3. Has the statistical analysis been performed appropriately and rigorously?

Reviewer #3: Yes

Reviewer #6: N/A

4. Have the authors made all data underlying the findings in their manuscript fully available (please refer to the Data Availability Statement at the start of the manuscript PDF file)?

Reviewer #3: Yes

Reviewer #6: Yes

5. Is the manuscript presented in an intelligible fashion and written in standard English?

PLOS Digital Health does not copyedit accepted manuscripts, so the language in submitted articles must be clear, correct, and unambiguous. Any typographical or grammatical errors should be corrected at revision, so please note any specific errors here.

Reviewer #3: Yes

Reviewer #6: Yes

Reviewer #3: Accepted this manuscript before

Reviewer #6: I don't know why this manuscript went for review again it should have been accepted in the last round

**Do you want your identity to be public for this peer review?** For information about this choice, including consent withdrawal, please see our Privacy Policy

Reviewer #3: **Yes:** Bala Nimmana

Reviewer #6: No
